# Computationally guided in-vitro vascular growth model reveals causal link between flow oscillations and disorganized neotissue

Eline E. van Haaften [1,2], Sjeng Quicken [3], Wouter Huberts [3], Carlijn V. C. Bouten [1,2✉] &
Nicholas A. Kurniawan [1,2]

Disturbed shear stress is thought to be the driving factor of neointimal hyperplasia in blood vessels and grafts, for example in hemodialysis conduits. Despite the common occurrence of neointimal hyperplasia, however, the mechanistic role of shear stress is unclear. This is especially problematic in the context of in situ scaffold-guided vascular regeneration, a process strongly driven by the scaffold mechanical environment. To address this issue, we herein introduce an integrated numerical-experimental approach to reconstruct the graft–host response and interrogate the mechanoregulation in dialysis grafts. Starting from patient data, we numerically analyze the biomechanics at the vein–graft anastomosis of a hemodialysis conduit. Using this biomechanical data, we show in an in vitro vascular growth model that oscillatory shear stress, in the presence of cyclic strain, favors neotissue development by reducing the secretion of remodeling markers by vascular cells and promoting the formation of a dense and disorganized collagen network. These findings identify scaffold-based shielding of cells from oscillatory shear stress as a potential handle to inhibit neointimal hyperplasia in grafts.

[1] Department of Biomedical Engineering, Eindhoven University of Technology, Eindhoven, The Netherlands. [2] Institute for Complex Molecular Systems (ICMS), Eindhoven University of Technology, Eindhoven, The Netherlands. [3] Department of Biomedical Engineering, CARIM School for Cardiovascular Diseases, Maastricht University, Maastricht, The Netherlands. ✉email: C.V.C.Bouten@tue.nl

A major clinical problem in hemodialysis therapy for end-stage renal-diseased patients is vascular access dysfunction. Vascular access for long-term hemodialysis is achieved through native arteriovenous (AV) fistula or synthetic arteriovenous graft (AVG). The primary cause of vascular access dysfunction in matured AV fistulas and AVGs is venous stenosis as a result of neointimal hyperplasia (NIH)[1]. The pathogenesis of venous NIH in vascular access is well described and thought to be initiated by vessel damage, graft bioincompatibility (for AVGs), uremia, and fluid wall shear stress[2]. These factors result in downstream effects such as oxidative stress, inflammation, and endothelial dysfunction, causing the migration and proliferation of smooth muscle cells to the intima.

High shear stresses (HSS) are persistently present in AVGs and AV fistulas, even years after vessel creation. Several studies have reported that regions with high wall shear stresses (in the range of 5–10 Pa) correspond to NIH formation[3–5]. On the other hand, Krishnamoorthy et al. has suggested an inverse correlation between wall shear stress (in the range of 5–80 Pa during peak flow) and stenosis formation[6]. It has also been proposed that low and oscillatory shear stress (in the range of <0.1–1 Pa) favors sites of stenosis[7] and that the temporal gradient of shear stress (in the range of +0.5 Pa/week up to 3.5 Pa) correlates with intimal medial thickening[8,9]. These seemingly inconsistent findings raise important questions about both the mechanistic foundation and the validity of the conjecture that wall shear stress stimulates the initiation and development of NIH in vascular access vessels. Indeed, systematic reviews reveal that the clinical evidence for this disturbed flow theory is weaker than generally believed and that so far very little is known about how altered flow is related to the cellular processes underlying NIH[10,11].

The need for mechanistic insights in (patho)physiological situations is especially critical with the emergence of a relatively new paradigm in regenerative medicine: in situ tissue engineering. This strategy hinges on the implantation of directly functional, bioresorbable, cell-free scaffolds, e.g., to be used as AVGs, which direct tissue regeneration at the locus of implantation to grow the neotissue into a state of mechanical homeostasis, presumably reducing the risk of NIH. The merit of the tissue engineering approach was recently highlighted in a study by Kirkton et al., where the authors implanted a bioengineered, acellular, human vessel as a hemodialysis conduit in patients with end-stage renal disease[12]. These vessels completely recellularized in this complex in vivo environment and transformed into a functional, living tissue, allowing repeated cannulation for years.

The new tissue has, so to say, "emerged" from its complex in vivo environment, whose convolution of individual factors, such as the interaction between cells and their 3D environment, paracrine, and juxtacrine signals from other cells, and response to mechanical forces, determined the resulting tissue properties. Intriguingly, these factors also affect the secretion of proteinases (e.g., MMPs and TIMPs), cytokines (e.g., MCP-1 and IL-6), and growth factors (e.g., TGF-$\beta$1), which are known to be involved in NIH formation[10]. To advance this promising therapeutic development, it is of paramount importance to dissect this complexity, because the knowledge about the role of each factor and the interplay between these factors provides crucial design parameters for guiding the developing tissue toward mechanical homeostasis[13].

In the present study, we introduce a numerical-experimental approach to systematically investigate how neotissue develops under the influence of the demanding hemodynamic environment in regenerating AVG scaffolds (Fig. 1). To quantitatively parameterize this environment, we use computational fluid dynamics (CFD) and fluid–structure interaction (FSI) modeling of a clinically derived vein–graft anastomosis. We then simulate this environment using our recently developed in vitro model

system[14]. This in vitro system is used to mimic the (early) pro-inflammatory stages of scaffold-driven in situ tissue formation, based on (1) the co-culture of human tissue-producing vascular cells and macrophages, which are (2) seeded in electrospun grafts, and (3) cultured in a bioreactor platform that allows for the independent control of wall shear stress and strain. This unique approach allows the delineation of the roles of biomechanical tissue environment and cellular responses, as well as their interplay, in scaffold-guided vascular tissue formation.

The findings revealed that various wall shear stress metrics (low, high, and oscillatory), in the presence of cyclic strain, differently regulate NIH- and tissue growth-related protein secretion, tissue growth, and remodeling. In particular, oscillatory shear stress promoted the formation of a dense and disorganized collagen network. Together, these insights confirm the causative relationship between different shear stress modes and vascular tissue growth and remodeling in 3D cyclically stretched constructs, and contribute to an improved understanding of scaffold-guided tissue regeneration and the initiation mechanism of NIH in vascular access vessels.

## Results and discussion

**Hemodynamics at the NIH-prone sites in the AVG.** In the early stages of in situ scaffold-guided vascular regeneration, the scaffold largely determines the mechanical performance of the overall construct. To quantitatively understand the hemodynamic environment in the NIH-prone sites of these scaffolds when implanted as AVG, we computed the local hemodynamics using numerical models (Fig. 2a). For this purpose, a realistic AVG geometry was reconstructed from 15 months postoperative computed tomography angiography (CTA) scans and 2-weeks preoperative ultrasound diameter measurements of a single renal-diseased patient. For detailed information regarding the model reconstruction and numerical simulations, the reader is referred to the "Methods".

The flow disturbances at the perianastomotic region at the venous site are known to cause the most common graft-related complications. These flow disturbances are associated with the local deformations and local shear stresses of the venous perianastomotic wall. To accurately estimate these local stresses and strains, we developed an FSI model of this region, which took into account the temporally varying flow-induced wall deformations as well as the consequent flow profiles. The graft (0.63 mm wall thickness[15]) and the vein (0.385 mm wall thickness[15–17]) were modeled as a Neo-Hookean material, with a Young's modulus of 1.5 MPa for the graft[15,18] and 0.455 MPa for the vein[15]. A CFD model of the full geometry with a rigid wall assumption was used to obtain proper boundary conditions for the FSI model and to estimate the wall shear stresses in the graft. At the inlet boundary of the CFD model, a Doppler ultrasound-based velocity profile was prescribed, whereas at the proximal venous outlet a zero-pressure boundary condition was prescribed[19]. To mimic the peripheral bed and collateral venous flow, a six elements lumped parameter model was coupled to the distal arterial and venous outlets.

These simulations revealed that the shear stresses have time-averaged values of around 5 Pa with a low oscillatory shear index (Fig. 2c). The strains around the venous anastomotic border are in the order of 1% with extremes up to 2% at the anastomosis (Fig. 2b). To the best of our knowledge, this is the first study that quantifies wall shear stresses and strains in AVGs, giving a unique insight into the biomechanical environment inside these AVGs.

**Integrating computational output into in vitro model.** The biomechanical environment in the AVGs, as quantified from these computational outputs, were mimicked in our in vitro platform to recapitulate the local hemodynamics in vascular

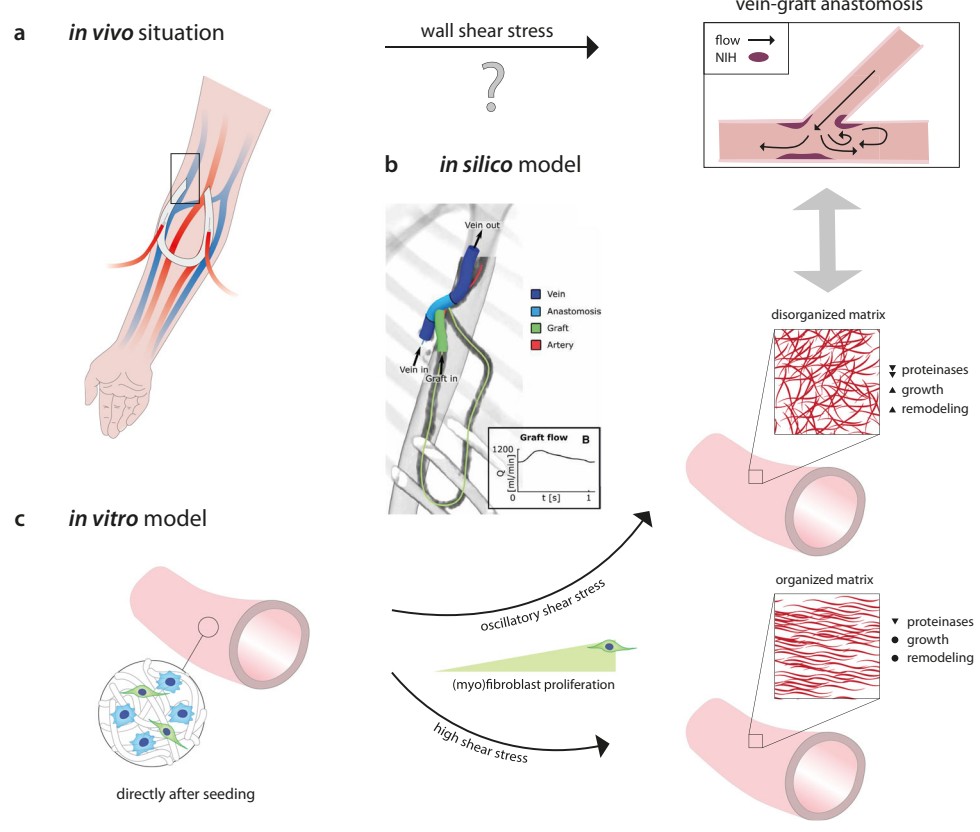

**Fig. 1 Proposed role of shear stress on in vivo neointimal hyperplasia, as revealed from a combined in silico and in vitro study. a** In vivo, neointimal hyperplasia at the venous anastomosis of vascular access grafts is a common occurrence. This pathology is presumably driven by shear stress. **b** Using in silico models (i.e., computational fluid dynamic (CFD) and fluid–structure interaction (FSI) models), the mechanical environment (i.e., the shear stresses and strains) in vascular access grafts is computed. **c** The results of these in silico models are used to define the boundary conditions for the in vitro model (i.e., a 3D scaffold containing macrophages and (myo)fibroblasts), in which the computed shear stresses and strains are recapitulated. After 14 days of dynamic culture, allowing for (myo)fibroblast proliferation and tissue formation, oscillatory shear stress resulted in a reduction of several proteinases and upregulation of growth and remodeling markers, which was accompanied by a dense and disorganized appositional matrix structure. This in contrast to high shear stress, which led to a strongly aligned appositional matrix. Note that in the in vitro study the shear stress and matrix alignment did not occur at the luminal side but at the outside of the scaffold. **b** is reproduced from Quicken et al.[15] with permission.

access vessels, i.e., the shear stresses and strains, allowing us to directly correlate to a biological response. This approach provides possibilities to test mechanistic hypotheses that are unaccessible in conventional in vivo studies, where the correlation between local hemodynamics, derived from computational models, and tissue composition, derived from histological stainings, is indirect and often only qualitative[9].

A selection of the computed hemodynamic parameters at the vein–graft anastomosis, representing a single snapshot of the spatiotemporal hemodynamic profile in this venous perianastomotic region, served as boundary conditions of the in vitro experiments (Fig. 2d). We selected a biomechanical condition that falls between the 15th and 100th percentiles of the strain and shear stress datasets (blue dotted lines in Fig. 2b (~2%), c (~3 Pa)), referred to as the HSS condition. Given the hypothesis that shear stress is the driving factor of NIH, we selected one other shear stress metric that is proposed to correlate with NIH: oscillating shear stress (OSS, ±3 Pa). To test the effects of HSS and OSS on neotissue development and NIH-involved protein secretion in our in vitro model of scaffold-guided vascular growth, we compared against a physiological (venous) low shear stress (LSS, ~0.5 Pa) control, as described by Malek et al[20].

The in vitro model consists of a cell-laden tubular scaffold construct mounted around impermeable silicone tubing and centered in a glass tube in a multi-cue bioreactor. The resulting annular channel is perfused with medium at a constant pressure gradient to apply laminar shear stress (in the HSS and LSS groups), or at an 1 Hz alternating pressure gradient to apply oscillatory shear stress (in the OSS group). The silicone tubing is cyclically pressurized up to a constant pressure at 1 Hz to apply circumferential stretch. For detailed description of the bioreactor, the reader is referred to our earlier work[21,22].

**The biomechanical environment during in vitro culture.** The vascular scaffolds for the in vitro experiments, with a wall thickness of 200 μm and inner diameter of 3 mm, were produced using electrospinning from poly(ε-caprolactone) bis-urea (PCL-BU, Fig. 3a). This polymer is soft, tough, biodegradable, and easy to functionalize, and therefore an attractive biomaterial for in situ tissue engineering[23]. The resulting scaffolds, with a Young's modulus of 3 MPa[14], exhibited an isotropic microstructure with ~5 μm fiber diameter (measured at the scaffold surface), which remained stable during the course of the culture, independent of the applied loading condition (Fig. 3b, c).

In addition to the fiber diameter, the morphology of the scaffold fibers remained unaffected (Fig. 3b), indicating that no scaffold degradation occurred at the scaffold surface. The degree of degradation was indeed expected to be minor in the present

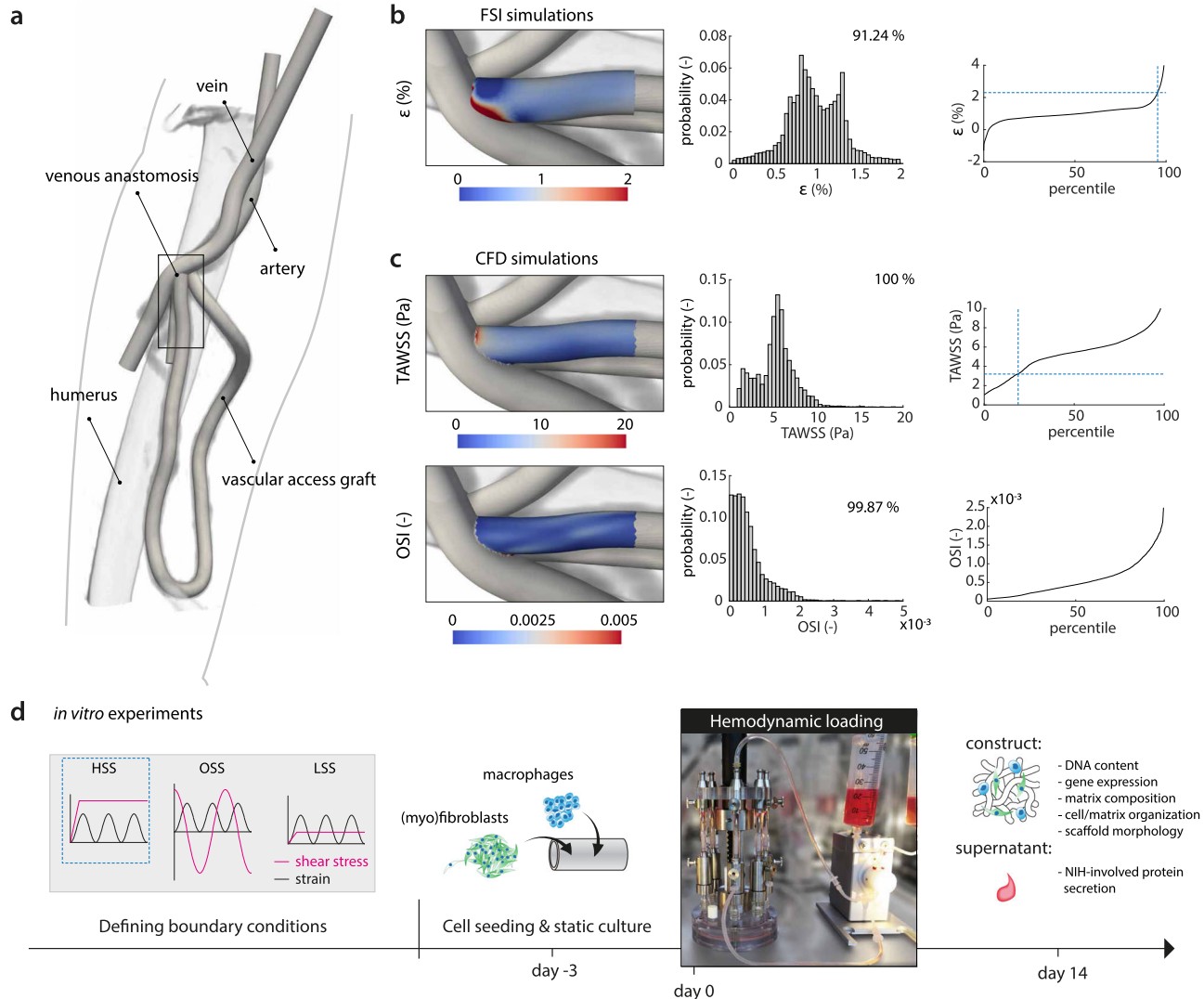

**Fig. 2 Computational characterization of the biomechanical environment in an AVG as input for scaffold-driven neotissue formation in vitro.**
**a** Geometry of the arteriovenous graft (AVG), looped between the axillary artery and axillary vein, that was used for the computational model (humerus inserted as a reference, venous anastomotic region indicated by rectangle). **b** Fluid–structure interaction (FSI) simulations and **c** computational fluid dynamics (CFD) simulations to compute the local strains and shear stresses in the AVG, respectively (histograms and associated percentile plots showing the distribution of the quantified read-out parameters ($\varepsilon$ strain, TAWSS time-averaged wall shear stress, OSI oscillatory shear index). The dashed lines in the percentile plots correspond to the boundary conditions of the high shear stress (HSS) group in the in vitro experiments. **d** The computational results guide the in vitro experiments by defining the relevant boundary conditions (HSS high shear stress, OSS oscillating shear stress, LSS low shear stress). (Myo)fibroblasts and PMA-stimulated THP-1 cells (macrophages) are co-seeded in the scaffolds using fibrin. After 3 days of static culture, the constructs are exposed to HSS, OSS, or LSS during 14 days, all in the presence of strain. After the dynamic culture, constructs and supernatants are collected and processed for further analysis. **a** is derived from Quicken et al.[19] with permission.

study, as we used ~10× lower cell seeding density compared to a previous study that investigated cell-induced scaffold degradation using THP-1-derived maraphages[24]. Future research can be further directed to examine degradation at the center of the scaffold, for example using Raman spectroscopy on cross-sections[25].

To mimic the early phase of the in situ scaffold environment, scaffolds were seeded with a 2:1 mixture of human THP-1-derived macrophages and primary vascular-derived (myo)fibroblasts (mFBs) using fibrin as a cell carrier. Following 3 days of static culture, the cell-seeded constructs were cultured for 14 days in Xanthan Gum (X-gum)-enriched medium containing L-ascorbic acid 2-phosphate (AA2P) under various shear stress conditions in the presence of low cyclic circumferential strain ($2.3 \pm 0.4\%$ at 1 Hz, Fig. 3d), mimicking the predicted strain

values in AVGs as computed by the FSI simulations (Fig. 2b). X-gum was added to match medium viscosity to blood viscosity (about $4 \cdot 10^{-3}$ Pa·s for a wall shear rate of ~200 s$^{-1}$)[26], while AA2P was added to stimulate matrix formation[27]. Rheology measurements indicated that the viscosity of the X-gum-enriched medium was ~2.5-fold higher compared to that of standard medium, confirming that medium viscosity was increased toward the range of blood viscosity (Fig. 3e)[26].

Samples in the HSS and OSS conditions were exposed to an average (absolute) shear rate of $1216 \pm 128$ s$^{-1}$, and samples in the LSS condition to $93 \pm 31$ s$^{-1}$ (left panels in Fig. 3f, g). The applied strains and shear rates were successfully maintained over the complete culture period, except at day 4 in the LSS condition where the shear rate dropped slightly to ≈60 s$^{-1}$ (left panel in Fig. 3g). Using the quantified viscosity of the X-gum-enriched

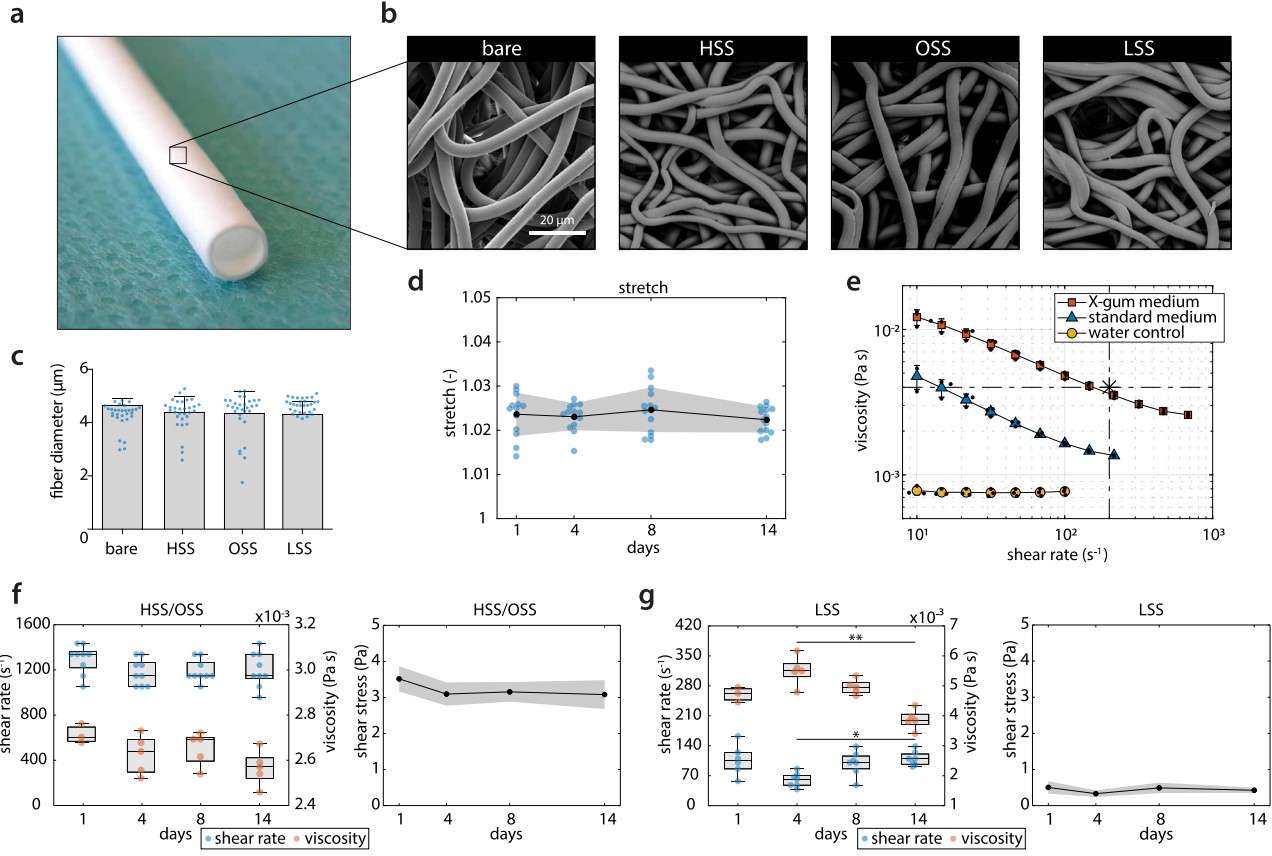

**Fig. 3 In vitro modeling of scaffold-guided vascular growth. a** Electrospun PCL-BU vascular scaffold prior to culture ($\phi$3 mm). **b** Scanning electron microscopy images of dynamically cultured samples after decellularization for each experimental condition, and of an 8-day statically cultured cell-free sample (bare). **c** Quantification of scaffold fiber diameter at the outside of the (decellularized) constructs ($n = 30$ fibers/condition). **d** Temporal variations in cyclic stretch for all loading conditions ($n = 13$ or 14/day). **e** Shear rate dependent viscosity in X-gum-enriched medium (red squares, $n = 3$), standard medium (blue triangles, $n = 3$), and water (yellow circles, $n = 3$). The * indicates blood viscosity at 200 s$^{-1}$. **f, g** Temporal variations in shear rate (left axis, $n = 6$, 7, or 9/day), viscosity (right axis, $n = 3$ or 5/day), and shear stress (right, derived from the actual (measured) medium viscosity) in the high, oscillating, and low shear stress conditions. Bars and black dots represent mean, error bars and gray area indicate standard deviation, boxplots contain 50% of the data with median highlighted by central mark (*$p < 0.05$, **$p < 0.01$).

medium, the shear rate applied to the samples translated to a shear stress of ($\pm$) 3.2 ± 0.3 Pa for the HSS (unidirectional) and OSS (complete flow reversal) condition (right panel in Fig. 3f), of which the magnitude is within the range of expected shear stress values in AVGs as computed by the CFD simulations (Fig. 2c). For the LSS condition, this translated to 0.44 ± 0.12 Pa (right panel in Fig. 3g), which is within the range of 0.1–0.6 Pa in healthy veins[20]. The applied shear stresses remained stable over time.

These results demonstrate that we were able to keep the biomechanical cellular environment, in terms of passive cues (via the scaffold fibers) and active cues (via the shear stresses and strains), stable and at the required level. This precise experimental control allowed us to directly correlate the biological response to the applied loading regime.

**Oscillatory shear stress activates macrophages and (myo)fibroblasts.** Using this dynamic in vitro setup, we quantified the cell and tissue growth and characterized the phenotypes of the cells in response to the different types of shear stress. The dynamic co-culture led to an overall increase of the construct mass with time, especially for the samples exposed to HSS and OSS (Fig. 4a). At day 14, scaffolds were completely populated with cells in all conditions and an increasing, but non-significant ($p = 0.143$), trend of cell content with high (81%) and OSS

(118%) compared to LSS was observed (Fig. 4b). However, the variation in DNA content within the groups was large, and no clear difference between the appearance and proliferative state of cells could be observed (Fig. 4c, d).

The appositional cell/matrix layer (i.e., the side that was exposed to the flow) aligned in the direction parallel to the flow (Fig. 4e, f). This layer contained elongated cells with stress fibers (top row in Fig. 4f). At the other side of the scaffold, cells with a more rounded morphology were observed (bottom row in Fig. 4f). Elongated cells stained positive for vimentin, indicating that the outside of the constructs was populated primarily by the mFBs (Fig. 5a). CD45, as an indicator of the macrophages, was mostly found in the middle layers of the constructs and to a lesser extent compared to vimentin. This was to be expected as a result of (myo)fibroblast proliferation and a possible reduction in macrophage number (e.g., as a result of apoptosis and transdifferentiation).

Previously, it has been shown in vivo that OSS ($\approx$14 Pa; range of 60 Pa) induces smooth muscle cell-rich plaque formation, while LSS ($\approx$10 Pa) induces the occurrence of M1-polarized macrophage-rich plaques in the carotid artery in mice[28–30]. Note that the seemingly large discrepancy between these magnitudes of wall shear stress and the definition of "LSS" we used in this study is attributable to the known inverse relationship between animal size and wall shear stress[31], and to the physiological difference in wall shear stress between arteries and veins. To test whether different biomechanical environments can indeed influence

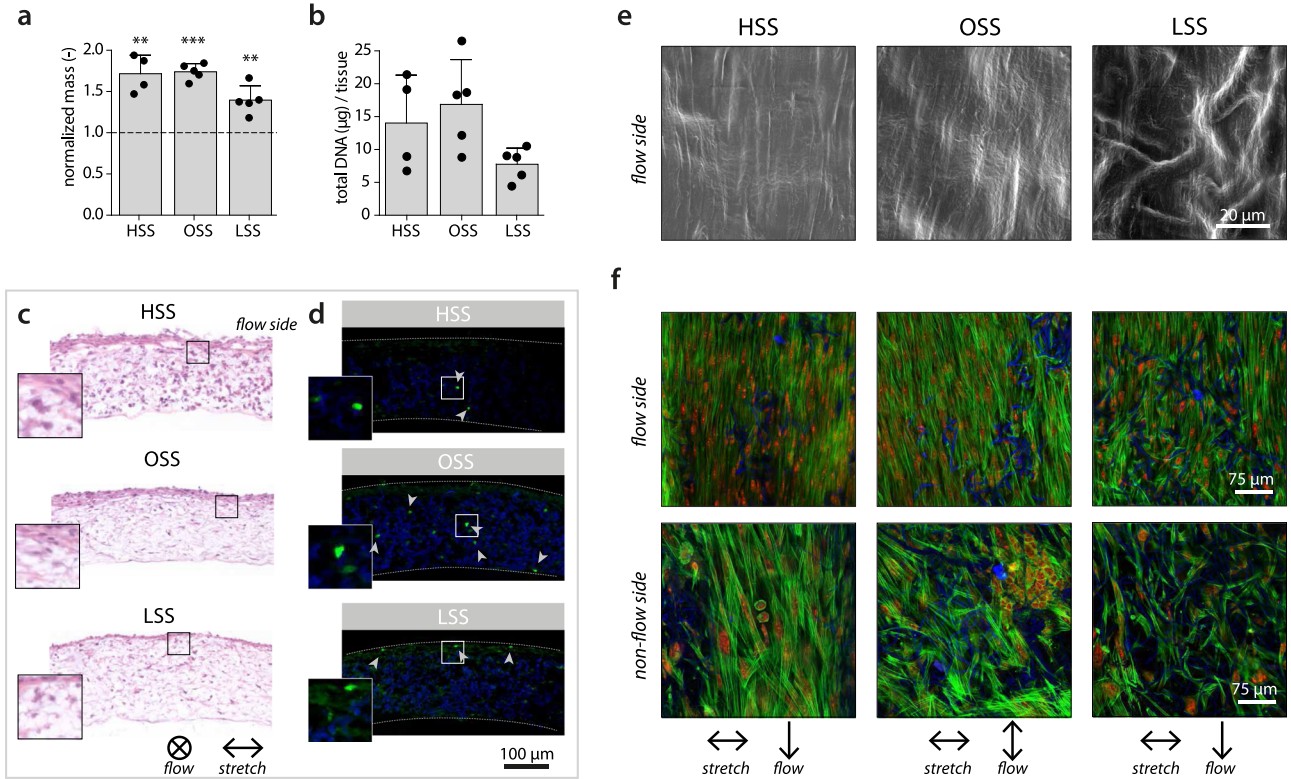

**Fig. 4 Cell and tissue growth at day 14 under different types of shear stress. a** Overall construct mass/surface at day 14 normalized to scaffold mass/ surface prior to seeding (dashed line indicates no change, **$p < 0.01$, ***$p < 0.0001$ tested with a one sample $t$-test). **b** Total DNA content per tissue construct. **c** Hematoxylin and eosin-stained cross-sections. **d** Localization of Ki67-positive cells (Ki67 in green, scaffold material in blue, dashed lines indicate construct borders). **e** Scanning electron microscope image of the cell and tissue morphology. **f** In-plane visualization of the F-actin cytoskeleton (actin in green, nuclei in red, and scaffold fibers in blue). Bars represent mean ± standard deviation. HSS high shear stress condition (3.2 Pa, $n = 4$), OSS oscillatory shear stress condition (±3.2 Pa, $n = 5$), LSS low shear stress condition (0.4 Pa, $n = 5$).

macrophage polarization, we examined the relative expression of M1 and M2 macrophage markers at HSS and OSS compared to LSS. Interestingly, we did not detect a polarization toward an M1 or M2 macrophage phenotype (Fig. 5b). Only *CD68* (pan-macrophage marker) and *IL10* (anti-inflammatory marker) were more clearly activated with OSS ($p = 0.0216$ and $p = 0.1224$, respectively), and to a lesser extent with HSS ($p = 0.1819$ and $p = 0.6740$, respectively, Fig. 5c). Similar to the dualistic macrophage phenotype, the phenotype of the mFBs could not be uniquely attributed to be either synthetic or contractile. Instead, we found simultaneous upregulation of *S100A4* (synthetic marker) and smoothelin (contractile marker), and downregulation of calponin (contractile marker) with OSS ($p = 0.0350$, $p = 0.0082$, and $p > 0.9999$, respectively) and HSS ($p = 0.1024$, $p = 0.8686$, and $p = 0.1096$, respectively, Fig. 5c).

Contrary to the gene-level analysis of MCP-1 ($p = 0.0075$), TNF-α ($p = 0.2423$), and IL-6 ($p = 0.1763$), at the protein level these cytokines were lower expressed in the OSS group compared to the LSS group, although the variation within each group was large (Supplementary Fig. S2). With respect to growth factors (TGF-β1 ($p = 0.1203$), CTGF ($p = 0.9973$)) and proteinases (MMP-1 ($p = 0.0122$), MMP-9 ($p = 0.0378$), TIMP-1 ($p = 0.0075$)), similar trends as seen for the cytokines were observed, which were significant for the proteinases, but not for the growth factors (Supplementary Fig. S2b). However, this quantification only represents a single snapshot in time, while similar work has shown that protein and RNA expressions are highly dynamic in nature[14]. In addition, the discrepancy between gene and protein secretion could indicate that the translation from RNA to protein is

differently regulated, or that the protein stability in the culture medium is reduced in HSS and OSS conditions[32].

Together, the results suggest that OSS activates mFBs and macrophages to grow new tissue in terms of both cell number and matrix. Moreover, the reduction of proteinase secretion in the OSS group suggests a suppressed tissue-remodeling environment, favoring neotissue formation rather than tissue degradation.

**Oscillatory shear stress promotes disorganized tissue growth.** Next, we sought to investigate the effect of the different shear stress metrics on tissue formation and remodeling. Overall, the gene expression of all growth and remodeling markers, with the exception of *TGFB1*, was (non-significantly) elevated with OSS compared to HSS (Fig. 6a). The expression of collagen type I/III, decorin, versican, fibrillin 1, *MMP1*, *MMP2*, and *TIMP1/2* was also higher compared to the LSS condition, which was statistically significant for decorin ($p = 0.0385$) and *MMP1* ($p = 0.0122$). Interestingly, with OSS the gene expression of *TIMP1* and *MMP9* was relatively high compared to the expression at the protein level (Supplementary Fig. S2b). In addition, *MMP1* and *MMP9* expression followed a similar trend at the protein level, but not at the gene level (Supplementary Fig. S2b), suggesting different posttranscriptional mechanisms between the loading conditions. TGF-β has been found to be an important posttranscriptional regulator of MMPs[33]. However, these mechanisms are complex and interrelated, complicating the comparison of the different MMPs, TIMPs, and growth factors to one another.

Collagen staining and hydroxyproline (HYP) quantification revealed that, compared to unidirectional shear stress (i.e., LSS

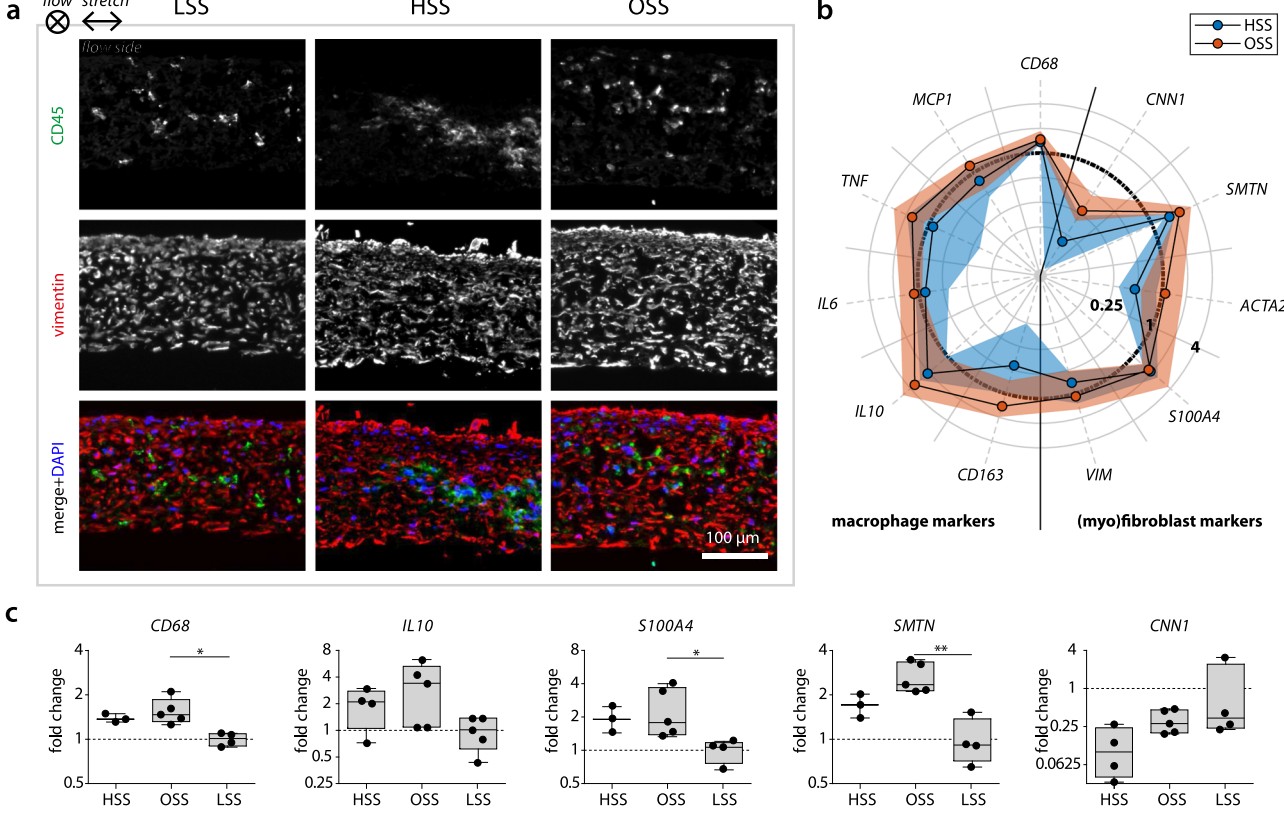

**Fig. 5 Phenotypical characterization of the cells in the vascular constructs at day 14. a** Representative cross-sections for each experimental group stained for CD45 (green) and vimentin (red) for the localization of macrophages and (myo)fibroblasts, respectively (nuclei are indicated in the lower panel in blue). **b** Relative gene expression compared to the low shear stress condition (indicated by the circle at 1) for macrophage-related genes (left part of the polar plot) and (myo)fibroblast-related genes (right part of the polar plot, $n \geq 3$/group). The dots and shaded areas indicate, respectively, the 50th and 25th–75th percentiles. **c** Boxplots for a selection of phenotypical-related genes ($n = 3$, 4, or 5, *$p < 0.05$, **$p < 0.01$). Boxplots contain 50% of the data with median highlighted by central mark. HSS high shear stress condition (3.2 Pa), OSS oscillatory shear stress condition (±3.2 Pa), LSS low shear stress condition (0.4 Pa). See Supplementary Fig. S1 for all boxplots of the gene expression data.

and HSS), OSS seemed to stimulate the synthesis of more and thicker collagen fibers (Fig. 6b, d). The differences in HYP content, however, are small and not significant. Collagen fibers are especially suited to resist in vivo loading and slowly take over the load-bearing properties of the resorbing scaffold. However, this process can result in scar-like tissue if collagen production happens too fast. HSS stimulated relatively more collagen type I formation while α-SMA-positive cells were predominantly detected at the tissue borders with LSS (Fig. 6c). Together, these observations are in line with the increased expression of growth markers at the gene level (Fig. 6a) and may have been further stimulated by the decreased levels of proteinases in the culture medium (Supplementary Fig. S2b).

We then focused on the neotissue organization in the appositional tissue layer which was in direct exposure to the shear stress and whose thickness was highest with HSS ($p = 0.0205$ compared to LSS, Fig. 6b, e). The collagen fibers were mainly oriented at an angle between 80° and 90° in all groups (i.e., parallel to the flow direction), and compared to LSS, consistently more aligned in the HSS condition, despite the limited differences in proteinase secretion (Supplementary Fig. S2b). In contrast, the peak in the OSS condition was much broader compared to LSS and HSS, indicating that the collagen is deposited with a substantially higher in-plane fiber dispersion when the flow is not unidirectionally applied. Based on these findings, we conclude that the collagen fiber dispersion in the superficial collagen layer is a direct result of flow-induced shear stress, rather than an indirect result of the secreted proteinases. On the other hand, the

overall collagen fiber orientation is not necessarily mediated by shear stress alone, as the presence of cyclic strain has also contributed to this fiber orientation[14].

In addition to tissue growth and remodeling by the mFBs, we aimed at scaffold degradation that is mainly induced by the macrophages. As the macrophages resided in the center of the scaffold and not at the surface, prohibiting the assessment of scaffold degradation by SEM, we examined the expression of genetic markers responsible for scaffold degradation. In the OSS group, *NFKB1* (a protein complex involved in oxidative stress) was upregulated compared to LSS (Supplementary Fig. S3, $p = 0.0044$). *LIPA* (lysosomal lipase) followed a similar trend. On the other hand, *NOX2* (ROS generating NADPH Oxidase 2 complex), which is involved in oxidative degradation, followed a decreasing, albeit non-significant, trend in the HSS condition. Although this could indicate enhanced oxidative and enzymatic degradative capacity of the macrophages with OSS, a previous study showed that the expression of these genes could not explain the differences in macrophage-driven scaffold mass loss, making it difficult to directly relate gene expression to scaffold degradation[24,34].

## Conclusion

This study aimed to elucidate how neotissue develops under the influence of the hemodynamic environment in regenerating AVGs via a combined in silico and in vitro approach (Fig. 1). With this approach, the hemodynamic environment could be

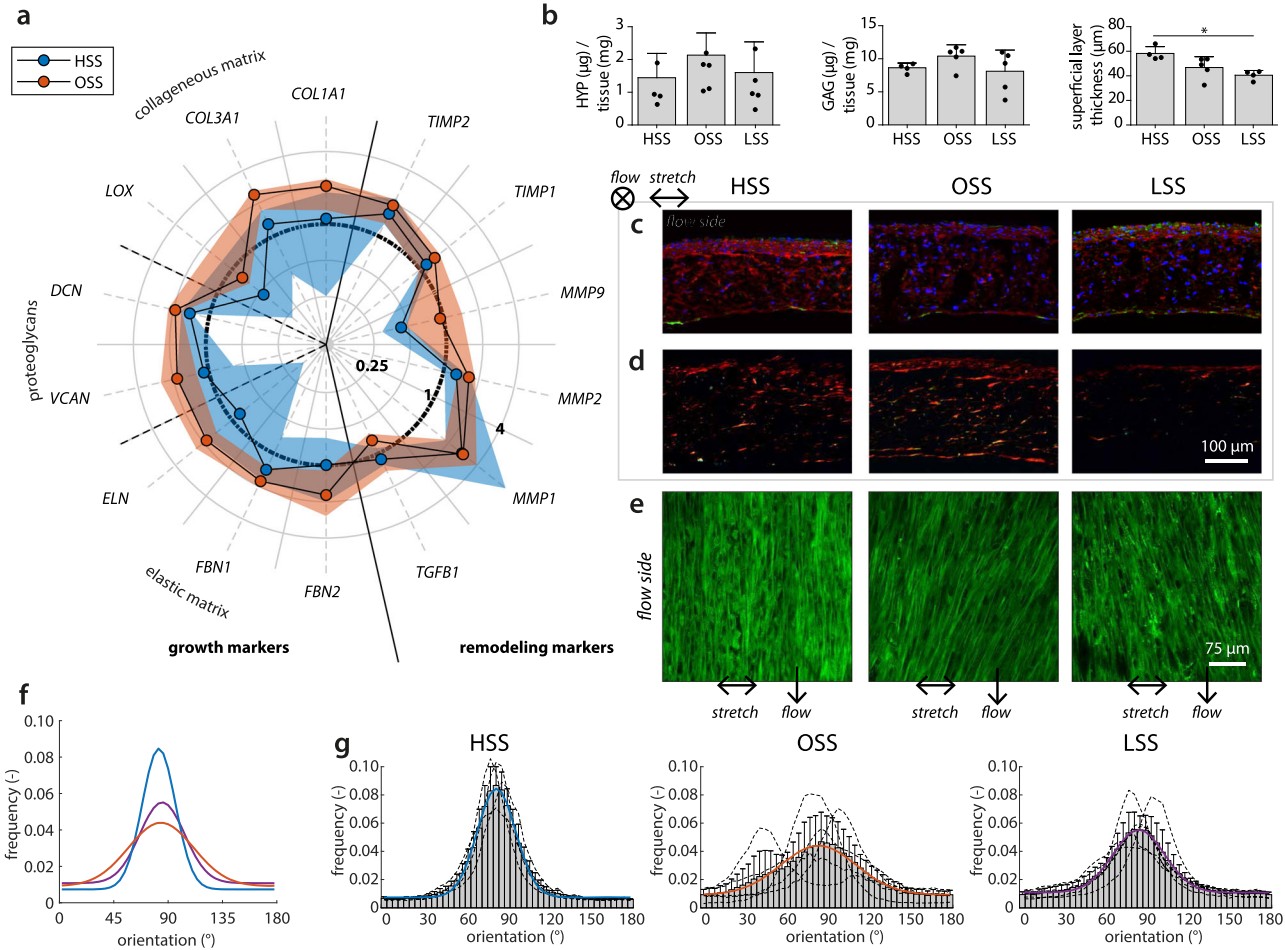

**Fig. 6 Changes to matrix growth, remodeling, and organization in response to shear stress. a** Relative gene expression compared to the low shear stress condition (indicated by the circle at 1) for markers related to growth (i.e., collageneous matrix, proteoglycan, and elastic matrix formation), as well as markers related to remodeling. The dots and shaded areas indicate, respectively, the 50th and 25th–75th percentiles. See Supplementary Fig. S1 for all boxplots of the gene expression data. **b** Total hydroxyproline (left) and glycosaminoglycan (middle) content normalized to total construct mass, and quantification of the thickness of the appositional collagen layer on top of the scaffold from the confocal z-stacks in **e** (right, *p < 0.05). Representative sections of each experimental group stained for (**c**) αSMA (green), collagen I (red), and DAPI (blue) and (**d**) Picrosirius red (visualized under polarized light). **e** Confocal visualization of the collagen structure. **f** Overlay plot and **g** separate histograms quantifying the angular collagen distribution in the appositional matrix layer for each condition (directions of applied shear stress (90°) and stretch (0°) indicated by the arrows in **c–e**. Bars represent mean ± standard deviation. HSS high shear stress condition (3.2 Pa, n = 4), OSS oscillatory shear stress condition (±3.2 Pa, n = 5), LSS low shear stress condition (0.4 Pa, n = 4 or 5).

controlled in a systematic way and biological responses could be directly measured, allowing the identification of causal relationships between hemodynamic parameters and the venous remodeling process. Relevant hemodynamic parameters were selected according to currently existing theories on venous remodeling (i.e., OSS and HSS[10]), while the exact magnitudes were derived from CFD and FSI simulations (Fig. 1a, b).

Using these in vitro dynamic co-cultures, we demonstrated that, compared to unidirectional shear stress (i.e., LSS and HSS), OSS activates both cell types to grow and remodel a tissue with a dense and disorganized structure (Fig. 1c). Furthermore, the secretion of NIH-related proteins was lower in the OSS and HSS conditions, indicating that the interaction between the different cell types and their mechanical environment resulted in a microenvironment that likely favored neotissue formation rather than tissue degradation. Here, it should be noted that other (complex) shear stress theories exist as well (e.g., spatial and/or temporal gradients in shear stress[10]), and that the cellular composition and phenotypes may differ in patients requiring vascular access, which should be the subject of future investigation.

Taken together, we identified oscillatory shear stress as a moderate, but progressive stimulator of cell proliferation and neotissue formation, even in the presence of cyclic strain. This is remarkable, because it was previously shown that laminar shear stress can have a stabilizing effect on strain-induced neotissue formation[14], making oscillatory shear stress a potential target to inhibit venous NIH and avoid excessive tissue formation. In the future, integration of these findings on cell mechano-response in the presence of complex hemodynamic situations into a computational framework of growth and remodeling can lead to the exciting possibility to predict the scaffold and venous (mal) adaptation to altered hemodynamics[35,36]. A thorough understanding of these mechanisms is essential to successfully translate in situ tissue engineering into a therapeutic approach as a solution to reduce vascular access dysfunction in patients with end-stage renal disease.

## Methods

**AVG geometry reconstruction.** Realistic axillary artery to axillary vein loop AVG geometries were used as input for the CFD and FSI models. These geometries were

reconstructed using clinical data of a single patient and assumed constant-diameter vessels. For the CFD AVG model, vessel paths and AVG configuration were assumed to remain relatively constant after AVG creation. As such, these AVG characteristics were extracted from a 15-month postoperative CTA scan by means of vessel-centerline extraction. Since vessel diameters may have changed significantly in the period between AVG creation and the CTA scan, 2-weeks preoperative ultrasound diameter measurements were used to estimate arterial and venous diameters (6.6 and 7.7 mm, respectively). Graft diameter was set to 6 mm. For the FSI model a similar approach was taken, though for this model zero-pressure vessel diameters needed to be imposed. The zero-pressure venous diameter was set to 7.1 mm, which was approximated using estimates of the average blood pressure at the time of ultrasound diameter measurements[15]. Zero-pressure graft diameter was set to 6 mm.

**Simulations**. The CFD equations were solved by using the OASIS[37] solver as implemented in the open-source finite element package FEniCS[38]. A mesh-independent solution was obtained at $3.1 \cdot 10^6$ tetrahedral Taylor-Hood elements and a time step of $1 \cdot 10^{-4}$ s. The FSI model was solved by using the Unicorn[39] solver that was also implemented in FEniCS. A mesh-independent solution was obtained at $2.5 \cdot 10^6$ and $0.6 \cdot 10^6$ linear tetrahedral elements in the fluid and solid domain, respectively. Blood was modeled as Newtonian fluid with a kinematic viscosity of $3.3 \cdot 10^{-6}$ m$^2$ s$^{-2}$ for the CFD simulations and $4.5 \cdot 10^{-6}$ m$^2$ s$^{-2}$ for the FSI simulations. From the CFD simulations, exposure to wall shear stress in the perianastomotic region was quantified using the time-averaged wall shear stress magnitude:

$$\mathrm{TAWSS} = \frac{1}{T} \int_0^T ||\vec{\tau}(t, \vec{x})|| \mathrm{dt} \qquad (1)$$

where $\vec{\tau}(t, \vec{x})$ represents the local WSS vector and $T$ the duration of the cardiac cycle. Exposure to oscillatory wall shear stress was assessed using the OSI:

$$\mathrm{OSI} = \frac{1}{2}\left(1 - \frac{||\int_0^T \vec{\tau}(t, \vec{x}) \mathrm{dt}||}{\int_0^T ||\vec{\tau}(t, \vec{x})|| \mathrm{dt}}\right) \qquad (2)$$

which ranges between 0 (unidirectional WSS) and 0.5 (purely oscillatory WSS). From the FSI simulations, wall strain was computed over the inner and outer surface of the graft wall according to:

$$\varepsilon = \sqrt{\frac{A_n - A_0}{A_0}} \qquad (3)$$

here, $A_0$ is the initial area of each surface element on the graft wall, whereas $A_n$ is the surface area of each element during the cardiac cycle. Finally, $\varepsilon$ was averaged over the complete cardiac cycle. The $\varepsilon$ strain metric is equivalent to engineering strain and combines both longitudinal and circumferential strain in a single scalar metric. For detailed information regarding the methodology of the simulations, the reader is referred to Quicken et al.[15,19].

**Scaffold preparation**. Tubular scaffolds ($\phi$3 mm, 2 cm in length, 200 µm wall thickness) were electrospun from PCL-BU (SyMO-Chem, Eindhoven, The Netherlands) in a climate-controlled cabinet (25 °C and 30% relative humidity; IME Technologies, Geldrop, The Netherlands). Briefly, 15% (w/w) PCL-BU was dissolved in 85% (w/w) CHCl$_3$ (Sigma, 372978) and delivered via a charged nozzle (−1 kV) at a flow rate of 40 µl min$^{-1}$ onto a positively charged (16 kV) rotating cylindrical mandrel ($\phi$3 mm, 500 rpm). The distance between the nozzle and the mandrel was kept constant at 16 cm. The resulting scaffolds were removed from the mandrel, dried in vacuo overnight, and placed over silicone tubing ($\phi$2.8 mm). The scaffold-wrapped silicone tubing was mounted in the bioreactor, sterilized by UV exposure (30 min/side), wetted in sterile H$_2$O, and incubated at 37 °C in complete medium (1:1 advanced Dulbecco's Modified Eagle Medium (a-DMEM):Roswell Park Memorial Institute 1640 (RPMI-1640) (Gibco, 124910 and A10491) with 10% fetal bovine serum (FBS; Greiner, Alphen aan den Rijn, The Netherlands), 1% penicillin/streptomycin (P/S; Lonza, DE17-602E), 0.5% GlutaMax (Gibco, 35050), and 0.25 mg ml$^{-1}$ AA2P (Sigma, A8960)) overnight to allow for protein adsorption.

**(Myo)fibroblast cell culture**. Vascular cells were isolated from surgical leftover of human vena saphena magna and used in accordance to the Dutch guidelines for secondary-use materials. The cells were expanded conforming conventional protocols[40]. Due to their structural properties, these cells have earlier been characterized as myofibroblasts[40,41]. However, a subpopulation of these cells do not express $\alpha$-SMA, which is more characteristic for fibroblasts[41]. Therefore, we refer to these cells as mFBs. mFBs were cultured in a-DMEM containing 10% FBS, 1% P/S, and 1% GlutaMax in a standard culture incubator (37 °C, 5% CO$_2$) and passaged at 80% confluency. Culture medium was changed every 3–4 days.

**THP-1 cell culture**. Human monocytic THP-1 cells (Sigma Aldrich, lot# 16K052) were cultured and expanded in suspension at a cell density of $(0.5–1.5) \cdot 10^6$ cells ml$^{-1}$ culture medium (RPMI-1640 containing 10% FBS and 1% P/S) in a standard culture

incubator (37 °C, 5% CO$_2$). Culture medium was changed three times per week. The cells tested negative on routinely performed (monthly) mycoplasm tests.

**Cell seeding**. After cell expansion, the mFBs (passage 6) and THP-1 cells (passage 10 after thawing) were seeded in the pre-wetted scaffolds using fibrin as a cell carrier[42]. Prior to seeding, the THP-1 cells were primed for 15 min in 50 ng ml$^{-1}$ phorbol 12-myristate 13-acetate (Sigma, P8139)-enriched culture medium to stimulate macrophage differentiation. To co-seed the cells into the porous scaffold, cells ($15 \cdot 10^6$ mFBs cm$^{-3}$ and $30 \cdot 10^6$ THP-1 cells cm$^{-3}$) were added to a mixture of bovine fibrinogen (10 mg ml$^{-1}$; Sigma, F8630) and bovine thrombin (10 IU ml$^{-1}$; Sigma, T4648), and carefully pipetted in two steps onto two opposing sides of the scaffold. Directly after pipetting the cell suspension, the cell-seeded scaffold was manually rotated until the suspension was completely absorbed by the scaffold[21,22]. To complete fibrin polymerization, the cell-seeded constructs were transferred to a 15 ml tube and kept in an incubator (37 °C, 5% CO$_2$) for 30 min, after which the tubes were filled with 8 ml of complete medium. To allow for cell adhesion to the scaffold fibers, the constructs were statically cultured in these tubes for 3 days prior to exposing the constructs to the different hemodynamic loading conditions (Fig. 2d).

**Shear stress and strain application**. After 3 days of static culture, the constructs were mounted in the culture chambers of the bioreactor for the application of hemodynamic loading[21,22]. To determine the loading conditions that mimic the conditions in vivo, we first examined the expected hemodynamic loads at the vein–graft anastomosis, as computed from the CFD and FSI simulations[15,19]. The samples were allocated to the different experimental conditions in a non-blinded, random way. The medium reservoirs were filled with 50 ml of complete medium, which was supplemented with 0.7 mg ml$^{-1}$ X-gum (Sigma, G1253) to increase the medium viscosity toward the range of blood viscosity[26]. To correct for possible medium evaporation during culture, the culture medium was supplemented up to 50 ml with sterile H$_2$O every 4 days, after which 25 ml of medium was replaced by fresh, X-gum-supplemented complete medium. After 14 days of dynamic culture, the tubular constructs were harvested, sectioned according to a cutting scheme (Supplementary Fig. S4), and stored at 4 °C (after 15 min fixing in 3.7% formaldehyde and 3 × 5 min washing in PBS) or −30 °C (after snap-freezing in liquid nitrogen) until further analysis. Supernatants from the culture medium after centrifugation ($300 \times g$, 5 min) at days 1, 4, 8, and 14 were stored at −30 °C until further analysis.

**Xanthan gum sterilization and dissolving**. In a sterile environment, X-gum was spread on a weighing paper in a large petri dish and UV-sterilized (10 min on two sides). After UV exposure, the X-gum was transferred to a pre-weighed 50 ml tube, and weighed again to determine the X-gum mass in a sterile way. Then, 70% EtOH was added at a concentration of 1.4 µl mg$^{-1}$ X-gum and the resulting suspension was transferred to 1:1 a-DMEM:RPMI-1640 at a concentration of 1.4 mg ml$^{-1}$. The sterilized X-gum was dissolved on a magnetic hot plate stirrer (4 h at 40 °C), resulting in a viscous solution. To prepare the X-gum-supplemented complete medium, this viscous solution was diluted 2× with 1:1 a-DMEM:RPMI-1640 to which the rest of the supplements were added (10% FBS, 1% P/S, 0.25 mg ml$^{-1}$ AA2P).

**Viscosity measurements**. To determine the shear stresses associated with the applied shear rates during the experiment, the viscous behavior of the supernatants (with and without X-gum) was quantified using a cone-plate rheometer ($\phi$50 mm) at 37 °C (ARES, Rheometric Scientific, 1 technical replicate/sample). The viscosity $\eta$ was measured at shear rates $\dot{\gamma}$ varying from 10 s$^{-1}$ until 1500 s$^{-1}$ (6 measurements per decade). The resulting viscosity–shear rate curves were linearly interpolated to determine the viscosities at low shear rate ($\dot{\gamma} \approx 100$ s$^{-1}$) and high shear rate ($\dot{\gamma} \approx 1200$ s$^{-1}$) (Matlab, The Mathworks, Natick, MA). The shear stress was calculated via: $\tau(\dot{\gamma}) = \dot{\gamma} \cdot \eta(\dot{\gamma})$.

**Biochemical assays**. The snap-frozen samples were used for quantification of DNA, glycosaminoglycan (GAG), and HYP content (two technical replicates/sample). After determination of the sample dry mass and surface (see "Determination of total mass"), samples were reduced to a fine powder using a micro-dismembrator (Sartorius) for sample digestion. Briefly, the samples were placed in cryovials containing 4 microbeads, frozen in liquid nitrogen, and disrupted at 3000 rpm for 60 s. To digest the sample, the powder was mixed with 500 µl digestion buffer (100 mM phosphate buffer (pH = 6.5), 5 mM L-cysteine (C-1276), 5 mM ethylenediamine tetraacetic acid (ED2SS), 140 µg ml$^{-1}$ papain (P4762); all from Sigma), transferred to a fresh Eppendorf tube, and incubated overnight at 60 °C. Prior to measurement of DNA, GAG, and HYP content, the samples were centrifuged at 12,000 rpm for 10 min. From the supernatant, DNA was quantified using the Qubit dsDNA BR assay kit (Life Technologies, Q32853) according to the manufacturer's protocol. GAG was quantified using a modified dimethyl methylene blue (DMMB)[43] assay with Shark chondroitin sulfate (Sigma, C4348) as a standard. Briefly, 40 µl of the supernatant and standards were mixed with 150 µl DMMB solution in a 96-well plate. The absorbance was measured using a microplate reader (540 nm, Synergy HTX, Biotek). HYP, as a measure of collagen, was quantified with a Chloramin-T assay[44] with trans-4-hydroxyproline as a reference (Sigma,

H5534). Before assaying, the samples were first hydrolyzed in 16 M sodium hydroxide (Merck, B1438798). The absorbance was measured using a microplate reader (550 nm, Synergy HTX, Biotek). DNA, GAG, and HYP values were normalized to the sample dry mass. Normalized DNA was multiplied by the sample total mass to obtain DNA content per sample (see "Determination of total mass").

**Determination of total mass**. To obtain the mass/surface, samples were weighed using a digital balance (XS105 dual-range analytical balance, Mettler Toledo, Switzerland) and photographed together with a ruler (one technical replicate/sample). The surfaces were measured from the photographs using ImageJ (v1.48, U.S. NIH, Bethesda, MD, USA). To estimate the increase in construct mass during culture, the mass/surface of the lyophilized samples at day 14 (see "Biochemical assays") was normalized to the initial mass/surface prior to seeding. To estimate the total sample mass at day 14, the mass/surface of the lyophilized samples was multiplied by the total surface of the samples (15 mm × 1.5$\pi$ mm).

**Scanning electron microscopy (SEM)**. Cell, tissue, and scaffold fiber morphology were assessed from SEM images (three predefined locations/sample). The formalin-fixed samples were placed in 0.25% glutaraldehyde (1 h), dehydrated in an ordered series of ethanol dilutions, and dried in vacuo overnight. After visualization of the cell and tissue morphology in low vacuum using a 10 kV electron beam (Quanta 600F, FEI, Hillsboro, OR, USA), half of the samples were decellularized in 4.6% sodium hypochlorite (15 min), washed in H$_2$O (2 × 5 min), and dried in vacuo overnight to visualize the scaffold fiber morphology. Together with the non-decellularized samples, samples were gold-sputtered and visualized in high vacuum using a 10 kV electron beam.

**Immunohistochemistry**. Cross-sections (7 μm cryosections from formalin-fixed samples) were washed in PBS (3 × 5 min), permeabilized in 0.5% Triton X-100 (30 min), and blocked for non-specific binding in 5% goat serum containing 1% BSA (30 min). The sections were then incubated with primary antihuman poly-clonal antibodies against Ki67 (rabbit IgG, 1:200, Thermoscientific, RB-1510-P0), CD45 (mouse IgG1, 1:1000, Abcam, ab33533), vimentin (mouse IgM, 1:2000, Abcam, ab20346), αSMA (rabbit IgG, 1:600, Abcam, ab5694), or collagen I (mouse IgG1, 1:200, Sigma, c2456), in 10× diluted block solution (overnight at 4 °C). After washing in PBS (3 × 5 min), the sections were incubated with 1:500 secondary goat antibodies labeled with Alexa-488 conjugate (antimouse IgG1 (for CD45, Molecular Probes, A21121) or antirabbit IgG (for αSMA, Molecular Probes, A11008)) or Alexa-647 conjugate (antimouse IgM (for vimentin, Jackson Immunoresearch, 115-605-075) or antimouse IgG1 (for collagen I and Ki67, Molecular Probes, A21240)), in 10× diluted block solution (60 min). Nuclei were stained with 4′,6-diamidino-2-phenylindole (1:500, Sigma). The stained sections were mounted in mowiol (Sigma, 81381) and visualized with an inverted epifluorescent microscope (Zeiss Axiovert 200M, ×20/0.5 Plan-Neofluar lens, three sections/sample).

**Fluorescence stainings**. Actin and collagen structures in formalin-fixed whole-mount samples were labeled with phalloidin-Atto 488 (1:200, Sigma) and CNA35-OG488 (1 μM[45]), respectively. Nuclei in the actin-stained samples were labeled with propidium iodide (7 μM, Molecular Probes, P3566). The stained samples were kept in PBS and visualized with a confocal laser scanning microscope (Leica TCS SP5X with a ×40/1.1 HCX PL Apo CS lens). The collagen orientation in each sample was quantified from two z-stacks (≈50 μm) recorded at two predefined locations using in-house developed software[21].

**Histological analysis**. Overall matrix content and fibrillar collagen in cross-sections (7 μm cryosections from formalin-fixed samples) were assessed from hematoxylin and eosin and Picrosirius red stains, respectively. Images were acquired with a bright field microscope (Zeiss Axio Observer Z1 with a ×20/0.8 Plan-Apochromat lens, three sections/sample). Picrosirius-stained sections were also imaged with polarized light to assess the birefringence of the collagen fibrils.

**Gene expression**. The snap-frozen samples were disrupted in a micro-dismembrator (see "Biochemical assays") and lysed in RLT lysis buffer (5 min, on ice). To isolate the RNA, the lysates were further processed using the Qiagen RNeasy kit with an additional DNAse treatment (30 min, Qiagen, 74106). RNA was eluted in 30 μl RNAse-free H$_2$O and quantified with a spectrophotometer (NanoDrop, ND-1000, Isogen Life Science, The Netherlands). Total RNA was reverse-transcribed into cDNA in a thermal cycler (protocol: 65 °C (5 min), on ice (2 min) while adding the enzyme mixture, 37 °C (2 min), 25 °C (1 min), 37 °C (50 min), and 70 °C (15 min)). The reaction solution consisted of 200 ng RNA, 1 μl dNTPs (10 mM, Invitrogen), 1 μl random primers (50 ng μl$^{-1}$, Promega, C1181), 2 μl 0.1M DTT, 4 μl 5× first strand buffer, and 1 μl M-MLV Reverse Transcriptase (200 U μl$^{-1}$, Invitrogen, 28025-013), which was supplemented to 20 μl with RNAse-free H$_2$O. qPCR was performed in a 10 μl reaction mix (two technical replicates/sample), containing 3 μl 100× diluted cDNA, 500 nM primer (forward and reversed, see Supplementary Table S1), and 5 μl SYBR Green Supermix

(Bio-Rad, 170-8886). Gene expression was normalized to GAPDH, identified as being most stable. $C_t$ values were acquired by exposing the reaction mixtures to the following thermal protocol: 95 °C (3 min), 40 cycles of 95 °C (20 s), 60 °C (20 s), and 72 °C (30 s), 95 °C (1 min), and 65 °C (1 min), concluded with a melting curve measurement. $C_t$ values were normalized for the housekeeping gene ($\Delta C_t$) and control ($\Delta\Delta C_t$, LSS condition), and the $2^{-\Delta\Delta C_t}$ formula was applied to calculate relative fold gene expressions of the genes listed in Supplementary Table S2. Datapoints were excluded if the melting curve was off, if >0.5 $C_t$ difference between technical replicates, or if the blanco was off in comparison to the measured $C_t$ value.

**ELISA**. Intimal hyperplasia-related proteins (Supplementary Table S2) were quantified from the supernatants at day 14 using a Luminex-based multiplex immunoassay (Multiplex core facility of the laboratory for Translational Immunology, UMC Utrecht, the Netherlands, one technical replicate/sample). In short, the supernatants were consecutively incubated with antibody-conjugated MagPlex microspheres (1 h), biotinylated antibodies (1 h), and streptavidin-phycoerythrin (10 min, diluted in high performance ELISA buffer (Sanquin)). Fluorescence intensity was measured using a FLEXMAP 3D system and analyzed by five-parametric curve fitting using Bio-Plex Manager software (Bio-Rad, version 6.1.1). Protein concentrations were normalized to the average DNA content per condition.

**Statistics and reproducibility**. To each loading condition in the culture experiment, $n = 5$ constructs from two separate experimental runs were assigned to allow for statistical analysis. No experimental-specific outcomes were observed. During the experiment, one sample from the HSS group dropped out due to experimental difficulties, resulting in $n = 4$ constructs for this group. Technical replicates were defined as repeated measurements of the same sample to assess the reproducibility of the analysis, and do not count to the total $n$. All data are presented either as mean ± standard deviation or as boxplots using Matlab (Matlab R2017; The Mathworks, Natick, MA) or Prism (GraphPad, La Jolla, CA, USA). Statistical analysis was performed to test for significant differences between the different shear stress modes with respect to biomechanical loading, biochemical content, gene expression, and cytokine secretion using a two-sided non-parametric Kruskal–Wallis test with a Dunn's multiple comparison test (GraphPad, La Jolla, CA, USA). Shear stress data were obtained by multiplying the daily average of the shear rate ($\bar{\gamma}$) with the daily average of the viscosity ($\bar{\eta}(\bar{\gamma})$). The standard deviation for shear stress was estimated based on error propagation. Gene expression data were logarithmically transformed prior to statistical analysis. Statistical significance was assumed for $p < 0.05$.

**Reporting summary**. Further information on research design is available in the Nature Research Reporting Summary linked to this article.

## Data availability
The source data underlying the graphs and charts presented in the main figures are available in the Supplementary Data file. The raw data are available from the corresponding author upon reasonable request.

## Code availability
All computational methods used in this paper to perform the CFD and FSI simulations were peer-reviewed and are extensively described in Quicken et al.[15,19]. The data and methods required to reproduce Fig. 2a–c from the simulation results can be obtained from a DOI-minting repository[46].

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

## Acknowledgements

The authors would like to thank Suzanne Koch and Tamar Wissing for their assistance in the in vitro experiments, Marina van Doeselaar and Marloes Janssen-van den Broek for their assistance in the sample analysis, and Anthal Smits for the fruitful discussions. This study is financially supported by ZonMw as part of the LSH 2Treat program (436001003) and the Dutch Kidney Foundation (14a2d507). N.A.K. acknowledges support from the European Research Council (851960). We gratefully acknowledge the Gravitation Program "Materials Driven Regeneration", funded by the Netherlands Organization for Scientific Research (024.003.013).

## Author contributions

C.V.C.B. and N.A.K. contributed equally. Author contributions according to the CRediT Taxonomy are as follows: conceptualization: all; formal analysis: E.E.v.H., C.V.C.B., and N.A.K.; investigation: E.E.v.H. and S.Q.; software: S.Q. and W.H.; funding acquisition: C.V.C.B.; supervision: N.A.K., C.V.C.B., and W.H.; visualization: E.E.v.H.; writing—original draft preparation: E.E.v.H.; writing—review and editing: all. All authors have approved the final article.

## Competing interests

N.A.K. is an Editorial Board Member for Communications Biology, but was not involved in the editorial review of, nor the decision to publish this article. Other authors declare no competing interests.
