## [Peer Review File · Communications Biology]

Reviewer comments:

Reviewer #3

(Remarks to the Author)

In the manuscript entitled "Computationally guided in-vitro vascular growth model reveals casual link between flow oscillations and disorganized neotissue", Van Haaften and coauthors highlight a trend topic in vascular tissue engineering, which concerns the neointimal hyperplasia that occurs in response to disturbed shear stress. They presented an integrated numerical-experimental approach to study how different mechanical stimuli can affect the neotissue formation in an in vitro vascular growth model. In my opinion, the quality of the presented data is high and the topic investigated could be of great interest for the readers of this Journal. The aims and the research structure are clear and well presented.

Specific minor comments:

1. The authors should better explain the seeding process. In particular, it is not clear how the seeding suspension (cells embedded in a fibrin gel) can colonize the totality of the scaffold. Is the scaffold subjected to a rotation during the seeding process or it is completely immersed in the fibrin gel?
2. For the preparation of SEM samples, authors reported a double fixation process (formaldehyde + glutaraldehyde). Could you explain the rationale of a double fixation?
3. In the materials and methods section the number of experiments and replicates for each test must be specified.

Reviewer #4

(Remarks to the Author)

Van Haaften et al. report here a numerical-experimental method to study the relation between shear stress and neotissue development and its relevance in the development of NIH. They first use a computational approach to quantify the hemodynamic environment and after they use these parameters to perform in vitro experiments mimicking the real conditions of the biological process. Although I found the study interesting, the manuscript has important weaknesses and I do not recommend the publication of the manuscript at this stage.

Major points:

In general, I found the figures very confusing, with the size of some panels not appropriated for the relevance of the panel, while some others that are less informative are very big. I also suggest to the authors to build a stronger discussion section where they can highlight their main achievements and the relevance of them to what is known so far. In the manuscript discussion is diluted among the results and sometimes seems weak, as well as highlighting the relevance and the novelty of their studies.

1)Line 82: The authors based all their computational calculations in a thickness of the vein of 0.385 mm. However, in reference 18 is reported a vein thicknesses of 0.51mm (which is about 40% higher) why are these values so different? I am afraid this really impacts the quantification.

2)In line 90 the authors reported an average shear stress value of 5 Pa and support this result with references 18 and 19. In reference 18 the reported values are about 10-15Pa and much lower below 2Pa in 19. Why the results obtained in this work are so different? Are they biologically relevant? Can the authors validate their estimations with experimental data?

3)About the experimental setup for the coculture. In scheme 1C it seems that cells are attached and grown in the inner part of the scaffold while in figure 1D it seems the cells are grown in the external part of the artificial scaffold this inaccuracy must be solved. Also the authors state in line 126-127 that the diameter of the fiber is about 5 μ m and it doesn't change even after 14 days of culture. Can they include how they estimate this value? I found very surprising that the

diameter is not affected even after 14 days of cell culture where cells are growing and secreting proteins and extracellular compounds that might attached to the scaffold.

4) In line 135, the authors provide the viscosity of the blood at a given shear stress of 200 s⁻¹. Can the authors include this value in fig 2E? This is key for the reader to properly judge if they are using the medium with a similar viscosity as they claim.

Also related to viscosity studies, fig 2D shows the dependence of viscosity with shear stress, however in panel G for day 4 the authors reported a drop in shear rate that do not affect significantly to the viscosity and in the day 14 despite there is no drop in shear stress the viscosity significantly drops. Can the authors discuss the reason for this?

In panel G right (as well as in panel F). The author report a stable shear stress and they claim the stability of their home made device from this data. Why then in panel G they performed a statistical analysis and show significant differences at day 4? is it relevant?

5) Line 163-164 "The appositional cell/matrix layer (i.e., the side that was exposed to the flow) aligned in the direction parallel to the flow (Fig. 3e)". SEM images have very poor resolution, even some orientation can be distinguished, the authors should provide a control image in which the cells have grown without any shear stress and also support this with fluorescence imaging.

6) In line 172-174, the authors discussed the influence of LSS in macrophage polarization. They referred to previous works that demonstrated macrophage polarization at LSS about 10Pa. After performing their test of macrophages polarization, they conclude that there is no polarization effect at their low LSS (0.5Pa). This fact raises to questions. Are they range of shear stress in these studies relevant to real physiological conditions and in the same range of previous studies? If so, why would one expect a polarization effect at a shear stress 20 times lower than the one reported to have an effect on cellular functions?

7) The authors report a discrepancy in the gene level of expression and protein expression of their biomarkers. They justify this through the "highly dynamic nature" of protein and RNA expression, the authors should either build a stronger discussion on this by providing references of similar works where such an effect was found and demonstrated to be due to the dynamics of RNA and protein expression or provide experimental data to support this.

8) Line 208-210. "Interestingly, the trends of MMP9 and TGFB1 are similar to the trends at the protein level (Fig. SIIIb), while MMP1 and TIMP1 follow an opposite trend, suggesting different mechanisms that regulate RNA to protein translation." The authors should provide experimental data that can support such a statement or appropriated references that can support this observation.

9) Line 211-212. "Collagen staining and HYP quantification revealed that, compared to unidirectional shear stress (i.e., LSS and HSS), OSS stimulated the synthesis of more and thicker collagen fibers (Fig. 5b, d)". Panel b shows not very clear differences and there is no statistical test done to show the significance of collagen changes at OSS and HSS and LSS conditions.

Minor points:

In line 18: The authors state that shear stress is one of the factors driving NIH development. Can the authors discuss what are the other factors and the relevance of shear stress vs the others?

Line 64-65: "for the first time the causative relationship between different shear stress modes and biological responses" This statement is too broad, there are many examples in literature showing the effect of shear stress in biological response. The authors should tempered the claims.

Line 157: "The dynamic co-culture led to an overall increase of the construct mass with time, especially for the samples exposed to HSS and OSS (Fig. 3a)." The authos should include an statistical test in this panel to properly support their claim.

Figure 3) I strongly recommend to the authors to change the way the labelled the panels rather than "*" using an explicit label indicating if cell have been or not under flow it is very helpful and

much more intuitive for the reader. Also using different thicknesses for the arrows to indicate different levels of shear rate is misleading and difficult for the comprehension of the paper. (Same for figures 5 and 6)

We thank the reviewers for their careful evaluation of the manuscript and the constructive comments. We have addressed all the comments, as outlined below, and made the corresponding changes in the revised manuscript.

1 Reviewer #1 (Remarks to the Author):

In the manuscript entitled “Computationally guided in-vitro vascular growth model reveals casual link between flow oscillations and disorganized neotissue”, Van Haften and coauthors highlight a trend topic in vascular tissue engineering, which concerns the neointimal hyperplasia that occurs in response to disturbed shear stress. They presented an integrated numerical-experimental approach to study how different mechanical stimuli can affect the neotissue formation in an in vitro vascular growth model. In my opinion, the quality of the presented data is high and the topic investigated could be of great interest for the readers of this Journal. The aims and the research structure are clear and well presented.

1.1 Specific minor comments:

1.1.1

The authors should better explain the seeding process. In particular, it is not clear how the seeding suspension (cells embedded in a fibrin gel) can colonize the totality of the scaffold. Is the scaffold subjected to a rotation during the seeding process or it is completely immersed in the fibrin gel?

The seeding process is described in detail in our previous works (van Haften et al. 2018, Koch et al. 2020). In particular, Koch et al. includes a written protocol of the seeding process and handling of the bioreactor, which will be further complemented with a video protocol later this year.

To briefly clarify the seeding process in the present manuscript, in the Methods section we refer to these earlier works and added a few extra details:

Methods:

IU·ml⁻¹; Sigma, T4648), and carefully pipetted in two steps onto two opposing sides of the scaffold. Directly after pipetting the cell suspension, the cell-seeded scaffold was manually rotated until the suspension was completely absorbed by the scaffold, as previously described [21, 22]. To complete fibrin polymerization, the cell-seeded constructs were transferred to a 15 ml tube and kept in an incubator (37 °C, 5 % CO₂) for 30 min, after which the tubes were filled with 8 ml of complete medium. To allow for cell adhesion to the scaffold fibers, the constructs were statically

Shear stress and strain application. After 3 days of static culture, the constructs were mounted in the culture chambers of the bioreactor for the application of hemodynamic loading as previously described [21, 22]. To determine

1.1.2

For the preparation of SEM samples, authors reported a double fixation process (formaldehyde + glutaraldehyde). Could you explain the rationale of a double fixation?

3.7%-Formaldehyde fixation was done directly at the end of the experiment for tissue samples reserved for SEM and stainings (see Supplementary Figure S1 for the cutting scheme). The glutaraldehyde step was done as part of the SEM preparation process, which was typically done a few days/weeks later, to ensure proper fixation.

We noted a small discrepancy, however, in the use of “formalin” and “formaldehyde” in the Methods sections describing the SEM and stainings. To be consistent, we replaced the word “formaldehyde” by “formalin” in the section describing SEM analysis:

Methods:

Scanning electron microscopy (SEM). Cell, tissue, and scaffold fiber morphology were assessed from SEM images. The formalin-fixed samples were placed in 0.25 % glutaraldehyde (1 h), dehydrated in an ordered series of

1.1.3

In the materials and methods section the number of experiments and replicates for each test must be specified.

We kindly refer the reviewer to the Methods section “Shear stress and strain application.”, in which we mention the total number of constructs assigned to each loading condition and describe the cutting scheme. As can be seen from the cutting scheme in Supplementary Figure S1, from each construct a sample was prepared for each analysis.

Depending on the analysis, we used single, duplicate or triplicate measurements per sample, as listed below:

Analysis	Sample obtained from	Group size	# technical replicates	Note
SEM	Construct	N=5 per loading condition (N=4 for HSS)	3	1 overview picture and 3 zoomed-in pictures per sample
qPCR	Construct	N=5 per loading condition (N=4 for HSS)	2	N/A
Stainings (cryosections)	Construct	N=5 per loading condition (N=4 for HSS)	3	3 sections per sample
Stainings (whole mount)	Construct	N=5 per loading condition (N=4 for HSS)	2	1 overview picture and 2 zoomed-in pictures per sample
Biochemical assays	Construct	N=5 per loading condition (N=4 for HSS)	2	N/A
ELISA	Medium	N=5 per loading condition (N=4 for HSS)	2	N/A
Viscosity	Medium	N=5 per time point	1	A selection of samples from different loading conditions were grouped together
Fiber analysis	Construct	N=1 per loading condition	30	Analysis on SEM pictures of 1 selected sample for decellularization per loading condition
Stretch	During culture	N=13-14 per time point	1	All samples from different loading conditions were grouped together
Flow	During culture	N=9 (HSS and OSI) and N=6-7 (LSS) per time point	1	All samples from HSS and OSI condition were grouped together.

We have added this extra information to the reporting summary. We have also added this information (in a simplified form) to the Methods.

Methods:

simulations [40, 1]. To each loading condition, $n = 5$ constructs from two separate experimental runs were assigned to allow for statistical analysis. The medium reservoirs were filled with 50 ml of complete medium, which was supplemented with $0.7 \text{ mg}\cdot\text{ml}^{-1}$ xanthan gum (XG; Sigma, G1253) to increase the medium viscosity towards the range of blood viscosity [26]. To correct for possible medium evaporation during culture, the culture medium was supplemented up to 50 ml with sterile H_2O every 4 days, after which 25 ml of medium was replaced by fresh, XG-supplemented complete medium. After 14 days of dynamic culture, the tubular constructs were harvested, sectioned according to a cutting scheme (Fig. SI), and stored at 4°C (after 15 min fixing in 3.7% formaldehyde and 3×5 min washing in PBS) or -30°C (after snap-freezing in liquid nitrogen) until further analysis. Supernatants from the culture medium after centrifugation (300 g, 5 min) at day 1, 4, 8, and 14 were stored at -30°C until further analysis. During the experiment, one sample from the HSS group dropped out due to experimental difficulties, resulting in $n = 4$ constructs for this group.

2 Reviewer #2 (Remarks to the Author):

Van Haften et al. report here a numerical-experimental method to study the relation between shear stress and neotissue development and its relevance in the development of NIH. They first use a

computational approach to quantify the hemodynamic environment and after they use these parameters to perform in vitro experiments mimicking the real conditions of the biological process. Although I found the study interesting, the manuscript has important weaknesses and I do not recommend the publication of the manuscript at this stage.

2.1 Major points:

In general, I found the figures very confusing, with the size of some panels not appropriated for the relevance of the panel, while some others that are less informative are very big. I also suggest to the authors to build a stronger discussion section where they can highlight their main achievements and the relevance of them to what is known so far. In the manuscript discussion is diluted among the results and sometimes seems weak, as well as highlighting the relevance and the novelty of their studies.

We thank the reviewer for the time and effort to critically review our manuscript and pointing out how to strengthen the message and Discussion. We would first like to reiterate the main findings and achievements of our study here:

- *Oscillatory shear stress favors neotissue formation by reducing the secretion of remodeling markers by vascular cells and promoting the formation of a dense and disorganized collagen network.*
- *This is the first study that quantifies strains and wall shear stresses in AVGs.*
- *The precise experimental control allowed us to directly correlate the biological response to the applied loading regime.*
- *Oscillatory shear stress activates macrophages and myofibroblasts.*
- *Oscillatory shear stress promotes disorganized tissue growth.*
- *Even in the presence of cyclic strain, oscillatory shear stress may be a potential target to inhibit NIH and avoid excessive tissue formation.*

These points have been highlighted in the Abstract and the Results & Discussion section of the manuscript. We agree with the reviewer that some figures and text could benefit from clarifications and further discussion. In particular:

- *Our figures have been updated (see major points 3, 4, 5 and minor points 3 and 4 below). We believe these updates lead to more clarity (especially regarding the viscosity plots).*
- *We added a stronger discussion for the computational part and the biochemical part (see major points 1, 2 and 7, 8 below).*
- *We restructured our manuscript and moved the discussion of scaffold degradation early in the Results & Discussion section (see major point 3 below). We believe this makes the discussion less diluted.*
- *We added the appropriate statistical tests and clarifications in the text (see major points 4, 6, 9 and minor point 3 below).*

We are confident that by addressing the comments raised by the reviewer, we have strengthened the discussion and overall quality of the manuscript.

2.1.1

Line 82: The authors based all their computational calculations in a thickness of the vein of 0.385 mm. However, in reference 18 is reported a vein thicknesses of 0.51mm (which is about 40% higher) why are these values so different? I am afraid this really impacts the quantification.

It is important to note that we used two computational models: an FSI model, including modeling of wall behavior, to compute wall strains, and a CFD model (without taking into account wall behavior), to compute wall shear stresses. As such, the venous wall thickness can only impact our quantification of strain. In addition, our model focuses on the biomechanical environment of the graft, and not of the vein. We will therefore limit the discussion to the effect of vein thickness on the strains in the graft in our FSI model.

Within that context, we expect that increasing the venous wall thickness by ~40% (i.e., from 0.385 mm to 0.51 mm) has negligible impact on graft straining, because the graft is less compliant compared to the vein (1.5 MPa Young's modulus and 0.63 mm thickness compared to 0.445 MPa Young's modulus and 0.385 mm thickness). Even if the venous wall thickness was increased to (for instance) 0.51 mm, the graft would still be much less compliant than the vein. Consequently, graft strain is primarily counteracted by its own stiffness, with only a minor effect from the vein.

Despite the expected limited impact of venous wall thickness on graft strain, we realized that our manuscript would benefit from a clearer elaboration on the choice of wall thickness. The vein thickness reported in reference 18 (McGah et al.) is, however, not representative for our model, because McGah et al. analyzes a matured (7.6 years post-operative) arteriovenous fistula, whereas we focus on a direct-postoperative arteriovenous graft. Our work should therefore be compared to other computational studies that focus on vascular access early after creation, such as Decorato et al. (2014) and Guess et al. (2017). These studies set the (cephalic) vein thickness to 0.4 mm, which supports our choice of venous wall thickness. We have added these references to the text:

Results & Discussion:

graft (0.63 mm wall thickness [1]) and the vein (0.385 mm wall thickness [1, 16, 17]) were modeled as a Neo-Hookean material, with a Young's modulus of 1.5 MPa for the graft [1, 18] and 0.455 MPa for the vein [1]. A CFD model of the full geometry with a rigid wall assumption was used to obtain proper boundary conditions for the FSI model and to estimate the wall shear stresses in the graft. At the inlet boundary of the CFD model, a Doppler ultrasound-based velocity profile was prescribed, whereas at the proximal venous outlet a zero-pressure boundary condition was prescribed [19]. To mimic the peripheral bed and collateral venous flow, a six-elements lumped parameter model was coupled to the distal arterial and venous outlets.

2.1.2

In line 90 the authors reported an average shear stress value of 5 Pa and support this result with references 18 and 19. In reference 18 the reported values are about 10-15Pa and mucho lower below 2Pa in 19. Why the results obtained in this work are so different? Are they biologically relevant? Can the authors validate their estimations with experimental data?

We thank the reviewer for pointing out this inaccuracy. The comparison with references 18 and 19 (Decorato et al. (2014) and McGah et al. (2014)) should not have been made, because the underlying flow fields are different. Whereas the referenced studies describe blood flow in an arteriovenous fistula (i.e., flow diverges from a common vessel into two daughter vessels), our study focuses on blood flow at the venous side of an arteriovenous graft (i.e., flow converges from two vessels into a common daughter vessel).

We have therefore removed the comparison from the text.

Results & Discussion:

These simulations revealed that the shear stresses have time-averaged values of around 5 Pa with a low oscillatory shear index (OSI, Fig. 1c). The strains around the venous anastomotic border are in the order of 1% with extremes up to 2% at the anastomosis (Fig. 1b). To the best of our knowledge, this is the first study that quantifies wall shear stresses and strains in AVGs, giving a unique insight into the biomechanical environment inside these AVGs.

Unfortunately, since most studies only focus on the WSS metrics in the vein, there is a lack of published information on the WSS in the graft near the venous anastomosis. In fact, this was the reason for generating this information ourselves. Since it is not trivial using experimental methods to assess time-dependent/disturbed WSS metrics under complex flow conditions at a high resolution, (validated) computational flow models are often used for this task (Ene-lordache & Remuzzi (2017)). In our study we also opted for this approach.

In our model we solve the 3D Navier-Stokes equations that govern fluid flow using a computational fluid dynamics (CFD) model. The specific CFD implementation that was used in our study has previously successfully been validated using a benchmark problem that was introduced by the U.S. Food and Drug Administration for the specific purpose of validating CFD codes (Bø et al. (2015)). Furthermore, the AVG modeling setup used in our study (i.e., the geometry and boundary conditions) has also been used in a previously peer-reviewed computational study (Quicken, Delhaas, et al. (2020)). Hence, we are confident that the results agree with the in-vivo situation.

Bø, J., Bergersen, A. W., Valen-Sendstad, K., & Mortensen, M. (2015). Implementation, verification and validation of large eddy simulation models in oasis. National Conference on Computational Mechanics, February, 1–23.

Decorato, I., Kharboutly, Z., Vassallo, T., Penrose, J., Legallais, C., & Salsac, A.-V. (2014). Numerical simulation of the fluid structure interactions in a compliant patient-specific arteriovenous fistula. *International Journal for Numerical Methods in Biomedical Engineering*, 30(2), 143–159. <https://doi.org/10.1002/cnm.2595>

Ene-lordache, B., & Remuzzi, A. (2017). Blood Flow in Idealized Vascular Access for Hemodialysis: A Review of Computational Studies. *Cardiovascular Engineering and Technology*, 8(3), 295–312. <https://doi.org/10.1007/s13239-017-0318-x>

Guess, W. P., Reddy, B. D., McBride, A., Spottiswoode, B., Downs, J., & Franz, T. (2017). Fluid-structure interaction modelling and stabilisation of a patient-specific arteriovenous access fistula. *ArXiv*, April. <https://doi.org/10.1007/s10237-017-0973-8>

Quicken, S., Delhaas, T., Mees, B. M. E., & Huberts, W. (2020). Haemodynamic optimisation of a dialysis graft design using a global optimisation approach. *International Journal for Numerical Methods in Biomedical Engineering*, April, 1–14. <https://doi.org/10.1002/cnm.3423>

2.1.3

About the experimental setup for the coculture. In scheme 1C it seems that cell are attached and grown in the inner part of the scaffold while in figure 1D it seems the cells are grown in the external part of the artificial scaffold this is inaccuracy must to be solved. Also the authors state in line 126-127 that the diameter of the fiber is about 5 μm and it doesn't change even after 14 days of culture. Can they include how they estimate this value? I found very surprising that the diameter is not affected even

after 14 days of cell culture where cells are growing and secreting proteins and extracellular compounds that might attached to the scaffold.

The cells were seeded onto the external part of the scaffold as indicated in Figure 1D, but because of its porosity, the cells will populate the entire scaffold in the course of several days (as is clear from the cross-sectional stainings in Fig 3, 4, and 5). To avoid confusion, we updated Scheme 1C with the zoom pointing to the external part.

Scheme 1:

We understand that the reviewer was surprised to read that the scaffold was unaffected by the cells, and we are glad to clarify our statement.

First, it is important to note that we characterized the scaffold using scanning electron microscopy, which only allowed us to assess the scaffold microstructure at the surface. The fiber diameter was measured from these SEM pictures, and by comparing it to a bare scaffold, we concluded that there was no change in fiber diameter (Fig 2b, c). We have added this important detail to the manuscript. In addition, we moved the discussion on characterizing scaffold degradation using Raman spectroscopy at the end of the manuscript to the same section to make the discussion less diluted, and emphasize that we only assessed scaffold degradation from the SEM analyses (i.e. the outer part of the scaffold)

Second, only minor degradation to the scaffold was expected based on our previous work on degradation (e.g., Wissing et al. (2020) seeded a THP-1 cell concentration of roughly 10 times the concentration that we used (250×10^6 cells/ml vs 30×10^6 cells/ml)). We have added this detail to the manuscript as well.

Results & Discussion:

an isotropic microstructure with $\sim 5 \mu\text{m}$ fiber diameter (measured at the scaffold surface), which remained stable during the course of the culture, independent of the applied loading condition (Fig. 2b, c).

In addition to the fiber diameter, the morphology of the scaffold fibers remained unaffected (Fig. 2b), indicating that no scaffold degradation occurred at the scaffold surface. The degree of degradation was indeed expected to be minor in the present study, as we used $\sim 10\times$ lower cell seeding density compared to a previous study that investigated cell-induced scaffold degradation using THP1-derived macrophages [24]. Future research can be further directed to examine degradation at the center of the scaffold, for example using Raman spectroscopy on cross-sections [25].

2.1.4

In line 135, the authors provide the viscosity of the blood at a given shear stress of 200 s^{-1} . Can the authors include this value in fig 2E? This is key for the reader to properly judge if they are using the medium with a similar viscosity as they claim.

We agree with the reviewer. We have added the viscosity value of blood at a shear rate of 200 s^{-1} in fig 2E and adjusted the figure caption accordingly.

Figure 2:

Caption Figure 2:

cyclic stretch for all loading conditions ($n = 13\text{--}14/\text{day}$). (e) Shear-rate dependent viscosity in XG-enriched medium (red squares, $n = 3$), standard medium (blue triangles, $n = 3$), and water (yellow circles, $n = 3$). The * indicates blood viscosity at 200 s^{-1} . (f, g) Temporal variations in shear rate (left, $n = 6\text{--}9/\text{day}$),

Also related to viscosity studies, fig 2D shows the dependence of viscosity with shear stress, however in panel G for day 4 the authors reported a drop in shear rate that do not affect significantly to the viscosity and in the day 14 despite there is no drop in shear stress the viscosity significantly drops. Can the authors discuss the reason for this?

The reviewer raises an interesting point that made us reconsider the way we presented the data obtained from the experiments. The shear rates were single values obtained during culture. In the original manuscript, we reported viscosity values at the average shear rate of the complete culture period. This explains why we did not observe a change in viscosity despite the drop in shear rate (note that one would expect an increase in viscosity, because of the shear-thinning behavior of the culture medium, as shown in Fig 2E). The shear stress values were then obtained as the multiplication of the average shear rate of the complete culture period with the viscosity values.

Thanks to the reviewer's comment, we now realize that this was not the most appropriate way to estimate the applied shear stresses. Thus, we redid the calculation. Viscosity is now reported as the daily viscosity value at the daily averaged shear rate. This accounts for the decrease in viscosity when the shear rate increases (i.e., shear-thinning behavior). Shear stress is then the multiplication of the daily averaged shear rate and the daily averaged viscosity. The standard deviations are calculated based on propagation of error.

Of note, this method result in an averaged shear stress that does not significantly vary over the entire culture period. However, because we now take into account the variation in both the shear rate measurement and the viscosity measurement, it led to a slight increase in the standard deviation.

We have updated Figure 2 and the text in the manuscript and Methods section accordingly.

Figure 2:

Caption Figure 2:

(f, g) Temporal variations in shear rate (left axis, $n = 6-9$ /day), viscosity (right axis, $n = 3-5$ /day), and shear stress (right, derived from the actual (measured) medium viscosity) in the high, oscillating, and low shear stress conditions. Bars and black dots represent mean, error bars and gray area

Results & Discussion:

Samples in the HSS and OSS conditions were exposed to an average (absolute) shear rate of $1216 \pm 128 \text{ s}^{-1}$, and samples in the LSS condition to $93 \pm 31 \text{ s}^{-1}$ (left panels in Fig. 2f, g). The applied strains and shear rates were successfully maintained over the complete culture period, except at day 4 in the LSS condition where the shear rate dropped slightly to $\approx 60 \text{ s}^{-1}$ (left panel in Fig. 2g). Using the quantified viscosity of the XG-enriched medium, the shear rate applied to the samples translated to a shear stress of (\pm) $3.2 \pm 0.3 \text{ Pa}$ for the HSS (unidirectional) and OSS (complete flow reversal) condition (right panel in Fig. 2f), of which the magnitude is within the range of expected shear stress values in AVGs as computed by the CFD simulations (Fig. 1c). For the LSS condition, this translated to $0.44 \pm 0.12 \text{ Pa}$ (right panel in Fig. 2g), which is within the range of $0.1 \text{ Pa} - 0.6 \text{ Pa}$ in healthy veins [21].

Methods:

Statistics. All data are presented either as mean \pm standard deviation or as boxplots. Statistical analysis was performed to test for significant differences between the different shear stress modes with respect to biomechanical loading, biochemical content, gene expression, and cytokine secretion using a two-sided non-parametric Kruskal-Wallis test with a Dunn's multiple comparison test (GraphPad, La Jolla, CA, USA). Shear stress data was obtained by multiplying the daily average of the shear rate ($\dot{\gamma}$) with the daily average of the viscosity ($\bar{\eta}(\dot{\gamma})$). The standard deviation for shear stress was estimated based on error propagation. Gene expression data was logarithmically transformed prior to statistical analysis. Statistical significance was assumed for $p < 0.05$.

In panel G right (as well as in panel F). The author report a stable shear stress and they claim the stability of their home made device from this data. Why then in panel G they performed a statistical analysis and show significant differences at day 4? is it relevant?

With the corrected shear stress calculation (see above), there is no statistical difference anymore between the time points. This can be explained by the shear-thinning behavior of the culture medium: significant changes in shear rate are compensated by opposite (significant) changes in viscosity, resulting in a stable shear stress.

We have therefore removed the statement from the text:

Results & Discussion

range of $0.1 \text{ Pa} - 0.6 \text{ Pa}$ in healthy veins [20]. The applied shear stresses remained stable over time, except for a small drop in the LSS condition at day 14.

2.1.5

Line 163-164 “The appositional cell/matrix layer (i.e., the side that was exposed to the flow) aligned in the direction parallel to the flow (Fig. 3e)”. SEM images have very poor resolution, even some orientation can be distinguished, the authors should provide a control image in which the cells have grown without any shear stress and also support this with fluorescence imaging.

In our experiments, we took into account a static sample, as shown below:

We agree with the reviewer that due to the lack of contrast, it is difficult to distinguish structural features on the SEM images. However, we do not think that the lack of contrast is an argument to include a control image of a sample grown without shear stress and stretch. In this study, we considered LSS as the “physiological, healthy control”. To avoid confusion for the reader, we decided not to include results of a static sample in the manuscript.

Instead of providing a control image we therefore propose to include higher contrast SEM images. The SEM images shown in the original manuscript are a combination of 2 channels (see pictures below). After inspection of the SEM image, we found out that due to the lack of contrast in Channel 2, the level of detail decreases in the combined image. We therefore decided to take Channel 1 only and updated Figure 3e accordingly. In addition, we slightly updated the text to indicate that the SEM images (cells + tissue) and fluorescence images (cells) both indicate the alignment in the direction of flow.

Figure 3e:

Results & Discussion:

The appositional cell/matrix layer (i.e., the side that was exposed to the flow) aligned in the direction parallel to the flow (Fig. 3e, f). This layer contained elongated cells with stress fibers (top row in Fig. 3f). At the other side of the scaffold, cells with a more rounded morphology were

2.1.6

In line 172-174, the authors discussed the influence of LSS in macrophage polarization. They referred to previous works that demonstrated macrophage polarization at LSS about 10Pa. After performing their test of macrophages polarization, they conclude that there is no polarization effect at their low LSS (0.5Pa). This fact raises to questions. Are they range of shear stress in these studies relevant to real physiological conditions and in the same range of previous studies? If so, why would one expect a polarization effect at a shear stress 20 times lower than the one reported to have an effect on cellular functions?

First of all, we considered the shear stresses present in normal veins as our physiological control (referred to in the manuscript as LSS), and evaluated the effect of HSS and OSS against this control. The range of magnitudes of the shear stresses applied in the present study is validated through our computational model and is representative of the expected in vivo range. Since HSS and OSS is normalized to LSS, the presented data neither can, nor was it our goal, to reveal anything about a polarization effect of LSS.

The main point we wished to make by referring to these previous studies, is to highlight that different modes of shear stress can have an effect on macrophage polarization. As we point out in the text, the range of shear stresses in the previous studies is different because these studies were performed in mice. In addition, these studies focus on arteries, whereas our study focusses on the graft anastomosis

to the vein. The cells from these studies are therefore 'primed' to a different shear stress type and magnitude compared to the cells from the present study.

To clarify this point, in the revised manuscript we have made it more explicit that our study focuses on the venous domain, whereas the studies we refer to focus on the arterial domain.

Results & Discussion:

To test the effects of HSS and OSS on neotissue development and NIH-involved protein secretion in our *in vitro* model of scaffold-guided vascular growth, we compared against a physiological (venous) low shear stress (LSS, ~ 0.5 Pa) control, as described by Malek *et al.* [21].

Results & Discussion:

rich plaques in the carotid artery in mice [28, 29, 30]. Note that the seemingly large discrepancy between these magnitudes of wall shear stress and the definition of 'low shear stress' we used in this study is attributable to the known inverse relationship between animal size and wall shear stress [31], and to the physiological difference in wall shear stress between arteries and veins. To test

2.1.7

The authors report a discrepancy in the gene level of expression and protein expression of their biomarkers. They justify this through the "highly dynamic nature" of protein and RNA expression, the authors should either build a stronger discussion on this by providing references of similar works where such an effect was found and demonstrated to be due to the dynamics of RNA and protein expression or provide experimental data to support this.

Generally, it is widely accepted that protein and RNA expressions vary dynamically. Nevertheless, we agree with the reviewer that our discussion can benefit from a solid evidence for this specific experimental system. We kindly refer the reviewer to our previous work (van Haaften & Wissing et al.(2020)) where we quantified protein and RNA expression at 2 different time points (day 3 and day 20) in a similar setup. The results presented in the paper indicate clear differences in the level of expression between the 2 time points. We have added this reference to the manuscript.

Results & Discussion:

which were significant for the proteinases, but not for the growth factors (Fig. SIIIb). However, this quantification only represents a single snapshot in time, while similar work has shown that protein and RNA expressions are highly dynamic in nature [15]. In addition, the discrepancy between gene and protein secretion could indicate that the translation from RNA to protein is differently regulated, or that the protein stability in the culture medium is reduced in HSS- and OSS-conditions [33].

2.1.8

Line 208-210. "Interestingly, the trends of MMP9 and TGFB1 are similar to the trends at the protein level (Fig. SIIIb), while MMP1 and TIMP1 follow an opposite trend, suggesting different mechanisms that regulate RNA to protein translation." The authors should provide experimental data that can support such a statement or appropriated references that can support this observation.

We thank the reviewer for pointing this out. Upon closer inspection of the data, the MMP9 expression at the gene level is not similar to the expression at the protein level. We have therefore rephrased the

results. In addition, because of the complex regulation mechanisms of MMPs and TIMPs, we have weakened our statement in the manuscript.

Results & Discussion:

Interestingly, with OSS the gene expression of TIMP1 and MMP9 is relatively high compared to the expression at the protein level (Fig. SIIIb). In addition, MMP1 and MMP9 expression follow a similar trend at the protein level, but not at the gene level (Fig. SIIIb), suggesting different post-transcriptional mechanisms between the loading conditions. TGF- β has been found to be an important post-transcriptional regulator of MMPs [33]. However, these mechanisms are complex and interrelated, complicating the comparison of the different MMPs, TIMPs, and growth factors to one another.

2.1.9

Line 211-212. “Collagen staining and HYP quantification revealed that, compared to unidirectional shear stress (i.e., LSS and HSS), OSS stimulated the synthesis of more and thicker collagen fibers (Fig. 5b, d)”. Panel b shows not very clear differences and there is no statistical test done to show the significance of collagen changes at OSS and HSS and LSS conditions.

The reviewer is correct that the differences in HYP content are not very clear. We have performed a statistical test (two-sided non-parametric Kruskal-Wallis test with a Dunn’s multiple comparison test) and could not detect significant differences. We have therefore weakened our statement.

Results & Discussion:

Collagen staining and HYP quantification revealed that, compared to unidirectional shear stress (i.e., LSS and HSS), OSS seemed to stimulate the synthesis of more and thicker collagen fibers (Fig. 5b, d). The differences in HYP content, however, are small and not significant. Collagen fibers are especially suited to resist *in vivo* loading and slowly take over the load-bearing properties of the resorbing scaffold. However, this process can result in scar-like tissue if collagen production happens too fast. HSS stimulated relatively more collagen type I formation while α -SMA-positive cells were predominantly detected at the tissue borders with LSS (Fig. 5c). Together, these observations are

2.2 Minor points:

2.2.1

In line 18: The authors state that shear stress is one of the factors driving NIH development. Can the authors discuss what are the other factors and the relevance of shear stress vs the others?

We updated the manuscript with a brief description of the other factors that are involved in NIH development. Since these “upstream factors” all contribute to multiple “downstream effects”, it remains speculative to position the relevance of shear stress against the other factors.

Introduction:

(NIH) [2]. The pathogenesis of venous NIH in vascular access is well described and thought to be initiated by vessel damage, graft bioincompatibility (for AVGs), uremia, and fluid wall shear stress (WSS) [3]. These factors result in downstream effects such as oxidative stress, inflammation, and endothelial dysfunction, causing the migration and proliferation of smooth muscle cells to the intima.

2.2.2

Line 64-65: “for the first time the causative relationship between different shear stress modes and biological responses” This statement is too broad, there are many examples in literature showing the effect of shear stress in biological response. The authors should tempered the claims.

This statement should be considered in the context of the study. What makes our study unique is the 3D aspect and the combination of shear stress with cyclic stretch, allowing the assessment of tissue growth and remodeling. We narrowed the statement in the text.

Introduction:

a dense and disorganized collagen network. Together, these insights confirm the causative relationship between different shear stress modes and vascular tissue growth and remodeling in 3D cyclically-stretched constructs, and contribute to an improved understanding of scaffold-guided tissue regeneration and the initiation mechanism of neointimal hyperplasia in vascular access vessels.

2.2.3

Line 157: “The dynamic co-culture led to an overall increase of the construct mass with time, especially for the samples exposed to HSS and OSS (Fig. 3a).” The authors should include an statistical test in this panel to properly support their claim.

The appropriate test to support our claim would be a comparison of the normalized mass to 1 using a one-sample t test. Although we cannot prove normality due to the small sample size, we opted to include this test, instead of the non-parametric version (Wilcoxon one-tailed signed rank test), because the non-parametric version is too conservative for the small sample size.

Figure 3:

Caption Figure 3:

fold mass/surface prior to seeding (*dashed line* indicates no change, ** $p < 0.01$, *** $p < 0.0001$ tested with a one-sample t test). (b) Total DNA content per tissue

2.2.4

Figure 3) I strongly recommend to the authors to change the way the labelled the panels rather than “*” using an explicit label indicating if cell have been or not under flow it is very helpful and much more intuitive for the reader. Also using different thicknesses for the arrows to indicate different levels of shear rate is misleading and difficults the comprehension of the paper. (Same for figures 5 and 6)

We have updated the labels in Figures 3, 4, and 5 to indicate the direction of the mechanical stimuli more explicitly. We kindly refer the reviewer to these updated figures in the revised manuscript.

REVIEWERS' COMMENTS:

Reviewer #3 (Remarks to the Author):

Second revision:

I appreciate the effort made by the authors to improve the paper and address all the comments. In my opinion, the manuscript is now improved, well structured and of great interest. All the minor suggestions reported in my previous revision have been addressed, and the additional details give much clarity to the methods involved in the study.

Reviewer #4 (Remarks to the Author):

The authors have made a big effort to properly discussed all my concerns. Below I just point out a minor comment, once this comment is addressed the manuscript can be accepted for publication.

Minor comments.

- In figure 3F, in the last row tagged as "non-flow side". Is the scale bar the same than the line above? A scale bar should be added or indicated in figure caption.

We thank the reviewers for their careful re-evaluation of the revised manuscript. We have addressed the minor comment as indicated by reviewer #4 and made the corresponding change in the revised manuscript. In this revision, we have also made the required changes to comply to the Final Revision Instructions.

1 Reviewer #3 (Remarks to the Author):

Second revision:

I appreciate the effort made by the authors to improve the paper and address all the comments. In my opinion, the manuscript is now improved, well structured and of great interest. All the minor suggestions reported in my previous revision have been addressed, and the additional details give much clarity to the methods involved in the study.

We thank the reviewer for the kind words.

2 Reviewer #4 (Remarks to the Author):

The authors have made a big effort to properly discussed all my concerns. Below I just point out a minor comment, once this comment is addressed the manuscript can be accepted for publication.

We thank the reviewer for the kind words.

2.1 Minor comments.

In figure 3F, in the last row tagged as “non-flow side”. Is the scale bar the same than the line above? A scale bar should be added or indicated in figure caption.

The author is correct in his/her assumption that the scale bar in the top row also applies to the bottom row. To make this unambiguously clear, we have added the scale bar also to the bottom row. In addition, we noticed that the scale bar in Fig. 4a was missing a label (i.e., 100 μm), which we now added.

Figure 3:

Figure 4: